# pMGF505-7R determines pathogenicity of African swine fever virus infection by inhibiting IL-1β and type I IFN production

Jiangnan Li☯, Jie Song☯, Li Kang, Li Huang, Shijun Zhou, Liang Hu, Jun Zheng, Changyao Li, Xianfeng Zhang, Xijun He(ORCID), Dongming Zhao(ORCID), Zhigao Bu*, Changjiang Weng(ORCID)*

Division of Fundamental Immunology, National African Swine Fever Para-reference Laboratory, State Key Laboratory of Veterinary Biotechnology, Harbin Veterinary Research Institute, Chinese Academy of Agricultural Sciences, Harbin, China

☯ These authors contributed equally to this work.
* buzhigao@caas.cn (ZB); wengchangjiang@caas.cn (CW)

**Data Availability Statement:** All relevant data are within the manuscript and its Supporting information files.

## Abstract

Inflammatory factors and type I interferons (IFNs) are key components of host antiviral innate immune responses, which can be released from the pathogen-infected macrophages. African swine fever virus (ASFV) has developed various strategies to evade host antiviral innate immune responses, including alteration of inflammatory responses and IFNs production. However, the molecular mechanism underlying inhibition of inflammatory responses and IFNs production by ASFV-encoded proteins has not been fully understood. Here we report that ASFV infection only induced low levels of IL-1β and type I IFNs in porcine alveolar macrophages (PAMs), even in the presence of strong inducers such as LPS and poly(dA:dT). Through further exploration, we found that several members of the multi-gene family 360 (MGF360) and MGF505 strongly inhibited IL-1β maturation and IFN-β promoter activation. Among them, pMGF505-7R had the strongest inhibitory effect. To verify the function of pMGF505-7R *in vivo*, a recombinant ASFV with deletion of the MGF505-7R gene (ASFV-Δ7R) was constructed and assessed. As we expected, ASFV-Δ7R infection induced higher levels of IL-1β and IFN-β compared with its parental ASFV HLJ/18 strain. ASFV infection-induced IL-1β production was then found to be dependent on TLRs/NF-κB signaling pathway and NLRP3 inflammasome. Furthermore, we demonstrated that pMGF505-7R interacted with IKKα in the IKK complex to inhibit NF-κB activation and bound to NLRP3 to inhibit inflammasome formation, leading to decreased IL-1β production. Moreover, we found that pMGF505-7R interacted with and inhibited the nuclear translocation of IRF3 to block type I IFN production. Importantly, the virulence of ASFV-Δ7R is reduced in piglets compared with its parental ASFV HLJ/18 strain, which may due to induction of higher IL-1β and type I IFN production *in vivo*. Our findings provide a new clue to understand the functions of ASFV-encoded pMGF505-7R and its role in viral infection-induced pathogenesis, which might help design antiviral agents or live attenuated vaccines to control ASF.

**Funding:** This study was supported by National Natural Science Foundation of China (31941002) (CW), National Natural Science Foundation of China (31872448) (JL), the State Key Laboratory of Veterinary Biotechnological Foundation (SKLVBP2018002) (JL), Major Scientific Research project of Chinese Academy of Agricultural Sciences (CAAS-ZDXT2018007) (CW), Natural Science Foundation of Heilongjiang Province of China (YQ2019C033) (JL). The funders had no role in study design, data collection and analysis, decision to publish, or preparation of the manuscript.

**Competing interests:** The authors have declared that no competing interests exist.

## Author summary

African swine fever virus (ASFV) causes a highly lethal swine disease that is currently present in many countries, severely affecting the pig industry. Despite extensive research, effective vaccines and antiviral strategies are still lacking and relevant gaps in knowledge of the fundamental biology of the viral infection cycle exist. In this study, we found that ASFV infection only induced low levels of IL-1β and type I IFNs in porcine alveolar macrophages (PAMs) and identified that pMGF505-7R, a member of the multigene family 505 (MGF505), strongly inhibited IL-1β and IFN-β production. ASFV lacking the MGF505-7R gene (ASFV-Δ7R) had reduced virulence in piglets and induced increased IL-1β and IFN-β production in PAMs and pigs compared with its parental ASFV HLJ/18 strain. Our results significantly increase our knowledge to understand functions of ASFV-encoded pMGF505-7R and its roles in pathogenesis, which may shed light on future research on live attenuated vaccines and antiviral strategies.

## Introduction

African swine fever (ASF) is one of the most lethal viral diseases affecting both domestic and wild swine. The acute infection of the ASF disease in domestic pigs leads to a 100% mortality rate with symptoms including high fever, vascular changes, cyanosis of the skin, abdominal pain, and diarrhoea [1]. Until now, there are no commercial vaccines and antiviral drugs available for ASF control. Therefore, the spread of ASF poses great economic losses to the pig industry and the ecosystems in the affected countries [2]. ASF virus (ASFV) is the etiological agent of the disease, and it is an enveloped, icosahedral, double-stranded DNA virus with a genome length ranging from 170 to 193 kb. ASFV has tropism for cells of the myeloid lineage, especially monocytes and macrophages, which seems to play a crucial role in viral infection-induced pathogenesis, viral persistence and dissemination [3,4]. It has been reported that more than 150 viral proteins are expressed in ASFV-infected macrophages, including proteins involved in the viral replication, virus-host interactions and modulation of host innate immune response [5].

  Previous studies documented that inflammatory factors are key components of host innate immunity, which pose a strong anti-viral effect against certain viruses such as adenovirus [6] and West Nile virus [7]. IL-1β, a potent pleiotropic proinflammatory cytokine produced predominantly by monocytes, macrophages, and lymphocytes, plays pivotal roles in regulating innate immune responses and instructing adaptive immune responses [8,9]. At least two separate signaling cascades are involved in the production and secretion of biologically active IL-1β. In the first cascade, pattern recognition receptors (PRRs) in host cells detect microorganisms, which induce transcription of the precursor pro-IL-1β [10]. In the second signaling cascade, the inflammasome activates the intracellular cysteine protease procaspase-1, which cleaves pro-IL-1β into mature IL-1β [10]. The NLRP3 inflammasome consists of NLRP3, adapter protein ASC and procaspase-1, and recognizes various types of exogenous and endogenous danger signals, including pathogens and endogenous damage-associated molecular patterns (DAMPs) [11]. A number of viruses are known to activate the NLRP3 inflammasome, including HCV, HIV and HSV-1 [12,13]. It has been reported that ASFV can induce IL-1β and TNF-α production in pigs with the increased inflammatory cytokines detected in the serum as early as 7 days post infection (dpi) [14]. One study showed that ASFV-encoded pL83L specifically interacts with IL-1β, and deletion of L83L gene did not affect the virulence

of the ASFV [15]. So far, there are no in-depth reports on the regulation of inflammatory responses by ASFV.

Type I IFNs are critical host immune mediators for suppressing the spread of viral infection. Cytoplasmic double-stranded DNA (dsDNA) can be sensed by cyclic GMP-AMP synthase (cGAS), which catalyzes the synthesis of cyclic GMP-AMP dinucleotide (cGAMP) [16]. cGAMP binds to and activates the stimulator of interferon genes (STING), which subsequently traffics from the endoplasmic reticulum to the trans-Golgi network where TBK1 is recruited and phosphorylated. This event allows the recruitment of IRF3 that is subsequently phosphorylated by TBK1 and then translocated to the nucleus to act as a transcription factor for type I IFNs [16]. A previous study showed that ASFV induces IFN-β production at 4 dpi in pigs [17]. ASFV NH/P68 strain can activate the cGAS-STING pathway in porcine alveolar macrophages (PAMs), while the virulent ASFV Armenia/07 strain suppresses the cGAS-STING pathway [18]. It has been reported that several genes of the virulent ASFV are involved in the inhibition of type I IFN production, such as genes belonging to the multigene family 360 (MGF360) and MGF505, and deletion of some of these genes results in attenuation of its virulence [19,20]. Importantly, attenuated ASFV strains which lack genes from MGF360 and MGF505 are more susceptible to type I IFNs [21]. Recently, a study shows that ASFV-encoded pMGF505-7R negatively regulates cGAS-STING Pathway by promoting ULK1-mediated STING degradation [22]. These results suggest that inhibition of the production of type I IFNs is an important strategy utilized by ASFV to evade host immune response and maintain infection. Thus, deciphering mechanism underlying inhibition of type I IFNs production by ASFV remains to be of great interest and importance for controlling ASF.

In this study, we demonstrated that ASFV infection only induced low levels of IL-1β and type I IFNs in PAMs, even in the presence of strong inducers such as LPS and poly(dA:dT), indicating that some of the ASFV-encoded proteins may inhibit IL-1β and type I IFN production. To explore which ASFV-encoded proteins were involved in suppressing IL-1β and IFN-β production, we first screened the members of MGF family and found that many members can suppress the IL-1β and IFN-β signaling. Among them, pMGF505-7R had the strongest inhibitory effect. Consistent with these results, a recombinant ASFV lacking the MGF505-7R gene (ASFV-Δ7R) induced much higher levels of IL-1β and IFN-β compared with its parental ASFV HLJ/18 strain. We also demonstrated that the higher induction of IL-1β by ASFV-Δ7R was dependent on TLR/MyD88 pathway and NLRP3 inflammasome. Mechanistically, pMGF505-7R interacted with IKK complex to inhibit pro-IL-1β transcription, and bound to NLRP3 to inhibit assembly of the NLRP3 inflammasome. Furthermore, we found that pMGF505-7R interacted with and blocked nuclear translocation of IRF3 to inhibit IFN-β production. Taken together, our findings reveal the functions of the viral pMGF505-7R on antagonizing host innate immune responses, which improve our understanding of ASFV pathogenesis and help design live attenuated vaccines or antiviral agents to control ASF.

## Results

### ASFV infection induces low levels of IL-1β and type I IFN in PAMs

The first ASF case was reported in China in August 2018, and then the disease rapidly spread across the entire country. An ASFV HLJ/18 strain, isolated from Heilongjiang (HLJ) province, is highly virulent and transmissible in domestic pigs [1]. To test whether the ASFV infection activates host innate immune responses, PAMs were mock-infected or infected with different doses of ASFV (multiplicity of infection (MOI) of 0.01, 0.1, or 1.0) for 24 h, and the secreted cytokines and its mRNA expressions in PAMs were detected by ELISA and qPCR, respectively. As shown in Fig 1A, the secretion and mRNA expression levels of IL-1β were both slightly

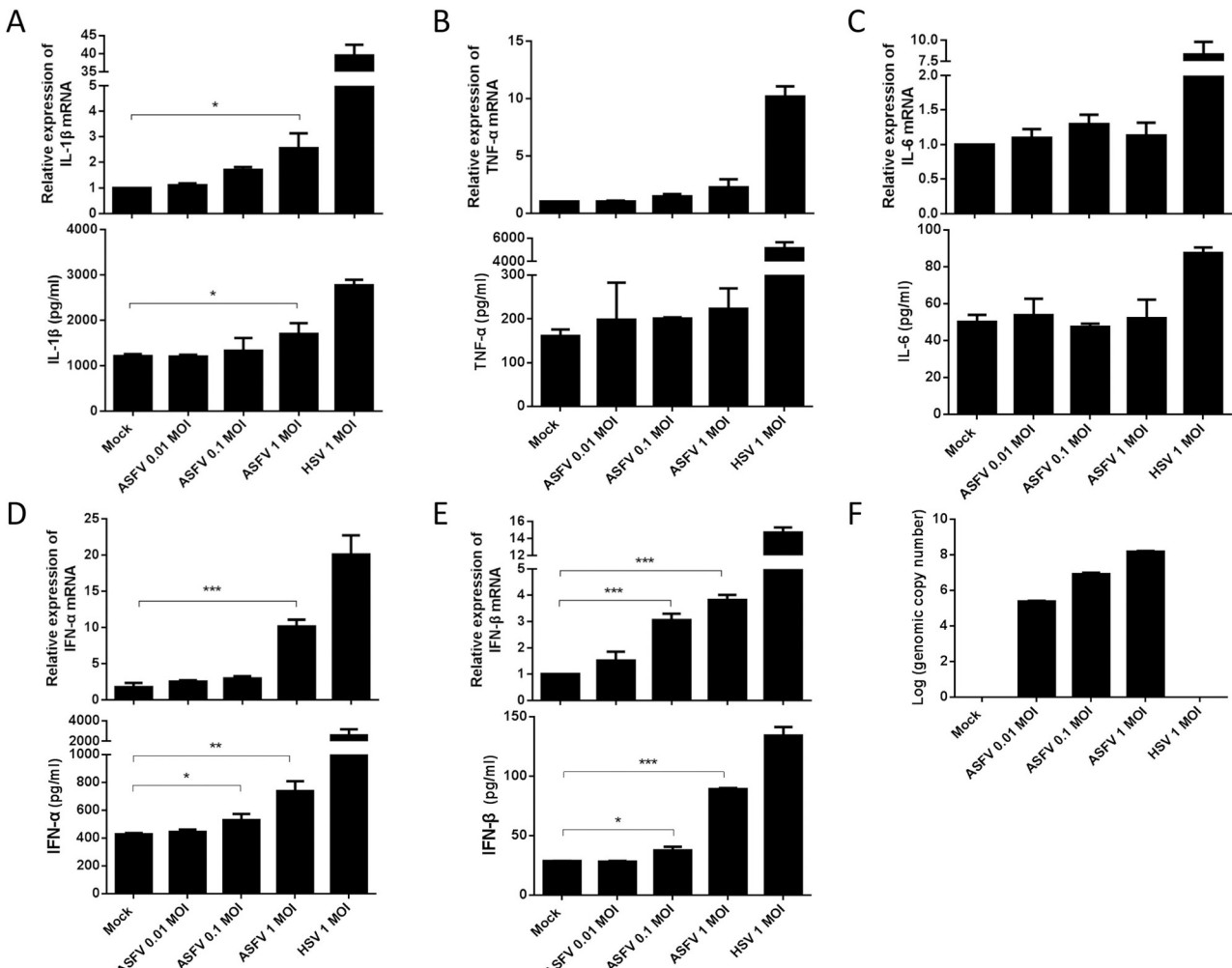

**Fig 1. ASFV infection induces low levels of IL-1β and type I IFN in PAMs. (A-E)** PAMs were either mock-infected or infected with ASFV at an MOI of 0.01, 0.1, and 1.0, respectively. At 24 hpi, the IL-1β (A), TNF-α (B), IL-6 (C), IFN-α (D) and IFN-β (E) levels in the cell culture supernatants were detected by ELISA and the mRNA levels in the cell lysates were determined by qPCR. PAMs were infected with HSV-1 at an MOI of 1.0 as a control. The genome copy numbers of ASFV in PAMs were measured by qPCR (F). A *p* value of less than 0.05 was considered statistically significant. $^{*}p<0.05$, $^{**}p<0.01$, $^{***}p<0.001$.

increased in a dose-dependent manner following ASFV infection, although secretion and mRNA expression levels of TNF-α and IL-6 were not obviously affected in PAMs (Fig 1B and 1C). Similarly, we found that ASFV infection also induced low levels of IFN-α and IFN-β in PAMs (Fig 1D and 1E). Herpes simplex virus type 1 (HSV-1), an internal positive control, induced significant IL-1β and type I IFN production as previous reported [23,24]. To confirm that the dose-dependent increases in IL-1β and type I IFNs production were actually due to increased infection of PAMs by ASFV, we tested the virus loads in PAMs following ASFV infection by qPCR, and found that more ASFV genome copies were detected in PAMs infected with higher MOIs of ASFV, indicating effective infection and replication of ASFV in PAMs (Fig 1F). Taken together, these results show that ASFV infection induces low levels of IL-1β and type I IFN in PAMs.

## ASFV infection inhibits inducer-mediated IL-1β and IFN-β production in PAMs

It has been reported that several ASFV-encoded proteins are involved in inhibition of type I IFN production [21], and our results indicated that ASFV infection induces very low levels of IL-1β and type I IFN production compared with HSV-1 (Fig 1). Therefore, we speculated that low levels of IL-1β and type I IFN production upon ASFV infection may be due to strong inhibition of these signaling pathways. To test this reasoning, PAMs were first infected with different doses of ASFV for 24 h, and then stimulated with strong proinflammatory and IFN inducers, such as LPS/Nigericin or LPS/poly(dA:dT), to induce these responses. We found that ASFV infection significantly inhibited LPS/Nigericin- and LPS/poly(dA:dT)-induced secretions and mRNA expressions of IL-1β in a dose-dependent manner (Fig 2A and 2B). Similarly, Sendai virus (SeV)- and poly(dA:dT)-induced secretions and expressions of IFN-β were also significantly suppressed in a dose-dependent manner by ASFV infection (Fig 2C and 2D). These results reveal that ASFV infection strongly inhibits inducer-mediated IL-1β and type I IFN production, proving that low levels of IL-1β and type I IFN inductions by ASFV infection result from strong suppression of these signaling pathways.

## Screening for ASFV-encoded pMGFs that inhibit IL-1β and IFN-β production

Since our results demonstrated that ASFV infection inhibits the IL-1β and type I IFN signaling pathways, we intended to screen the ASFV-encoded proteins that are responsible for the inhibition. ASFV is a large cytoplasmic DNA virus whose genome encodes more than 150 proteins. Several studies reported that some of the members of ASFV MGF360 and MGF505 were involved in the suppression of the host antiviral responses, and determination of viral virulence [19,25]. To test which members of ASFV pMGFs regulate IL-1β and IFN-β production, HEK293T cells were transiently transfected with the iGLuc-based inflammation reconstruction system reporter or IFN-β reporter and a plasmid encoding one of the fifteen pMGFs. We found that most members of pMGFs had inhibitory effects on both the iGluc reporter and the IFN-β reporter. Except two members of pMGF360s (18R and 19R), other members of pMGFs showed significant inhibitions of IL-1β maturation (Fig 3A). Among them, pMGF505-2R, pMGF505-7R and pMGF505-9R had the strongest inhibitory effects, especially pMGF505-7R. For IFN-β reporter screening, almost all of the pMGF360 and pMGF505 members showed significant inhibitions of IFN-β reporter activation (Fig 3B). We noticed that the inhibitory effect of pMGF505-7R was the strongest on both reporters. The expressions of pMGF members were confirmed by immunofluorescence assay (IFA) and Western blotting shown in S1 Fig. Based on these results, we focused on pMGF505-7R for further investigation of the mechanism underlying its inhibition of IL-1β and type I IFN production.

## ASFV pMGF505-7R inhibits IL-1β and IFN-β production

The production of IL-1β requires at least two separate signaling cascades: pro-IL-1β production and IL-1β maturation. It has been shown that activation of nuclear factor kappa B (NF-κB) signaling pathway is linked to the expression of pro-inflammatory cytokines, including pro-IL-1β [26]. To confirm the inhibition of pMGF505-7R on IL-1β production, HEK293T cells were transfected with the iGLuc-based inflammation reconstruction system or NF-κB reporter in combination with different amounts of pMGF505-7R, and then the luciferase signals were measured. As shown in Fig 4A and 4B, pMGF505-7R inhibited both the IL-1β maturation and NF-κB activation in a dose-dependent manner. We noted that levels of the

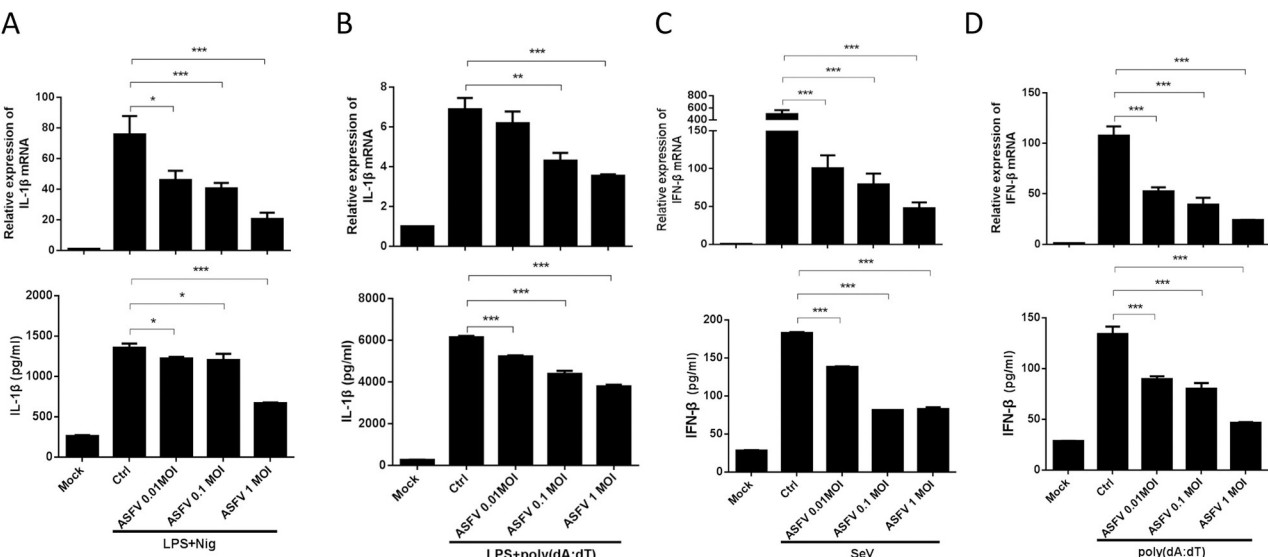

**Fig 2. ASFV infection inhibits inducer-mediated IL-1β and IFN-β production in PAMs. (A-D)** PAMs were either mock-infected or infected with ASFV at an MOI of 0.01, 0.1, and 1.0, respectively. At 24 hpi, PAMs were treated with LPS (100 ng/ml) for 8 h and then stimulated with Nigericin (5 μM) for another 4 h (**A**), or with LPS (100 ng/ml) and poly(dA:dT) (5 μg/ml) for 12 h (**B**), or with SeV (1 MOI) (**C**) or poly(dA:dT) (5 μg/ml) (**D**) for 12 h, then the secretion and the mRNA expression levels of IL-1β were detected by ELISA and qPCR, respectively. A *p* value of less than 0.05 was considered statistically significant. *$p < 0.05$, **$p < 0.01$, ***$p < 0.001$.

maturation form of IL-1β (p17) gradually decreased in the presence of increasing pMGF505-7R, which is consistent with the decreasing iGluc luciferase signals and demonstrates the gradual inhibition of IL-1β maturation. Similarly, to confirm the inhibition of pMGF505-7R on type I IFN production and function, HEK293T cells were transfected with the IFN-α, IFN-β, IFN-sensitive response element (ISRE) or IFN-stimulated gene 56 (ISG56) reporter along with increasing amounts of pMGF505-7R. The results showed that pMGF505-7R also dose-dependently inhibited activities of all of these reporters (Fig 4C–4F), demonstrating strong suppression of type I IFN signaling by pMGF505-7R. Taken together, our findings reveal that pMGF505-7R inhibits both IL-1β and type I IFN signaling pathways.

## ASFV-Δ7R induces higher IL-1β and type I IFN production compared with ASFV-WT

To determine the role of pMGF505-7R during ASFV infection, a recombinant ASFV lacking the MGF505-7R gene (ASFV-Δ7R) was generated from the highly pathogenic ASFV HLJ/18 strain by homologous recombination. The MGF505-7R gene was replaced by a cassette containing the fluorescent gene EGFP under the control of ASFV p72 promoter (S2A Fig). The recombinant ASFV-Δ7R was selected after 10 rounds of plaque purification in PAMs based on GFP expression (S2B Fig). To evaluate the accuracy of the genetic modification, the DNA sequence covering the modified region was amplified and sequenced, and the results showed successful replacement of MGF505-7R by p72-EGFP cassette (S2C Fig and S2 Table). The growth characteristics of ASFV-Δ7R *in vitro* were evaluated in PAMs. As shown in S2D Fig, ASFV-Δ7R displayed a similar growth kinetic compared to its parental ASFV HLJ/18 (ASFV-WT). Furthermore, We analyzed the transcriptome of viral genes in PAMs infected with ASFV-Δ7R or ASFV-WT, and found that except for MGF505-7R, there was no significant difference in the expression levels of other genes in MGF360 and MGF505 families in PAMs infected with ASFV-Δ7R compared with that infected with ASFV-WT (S3 Fig).

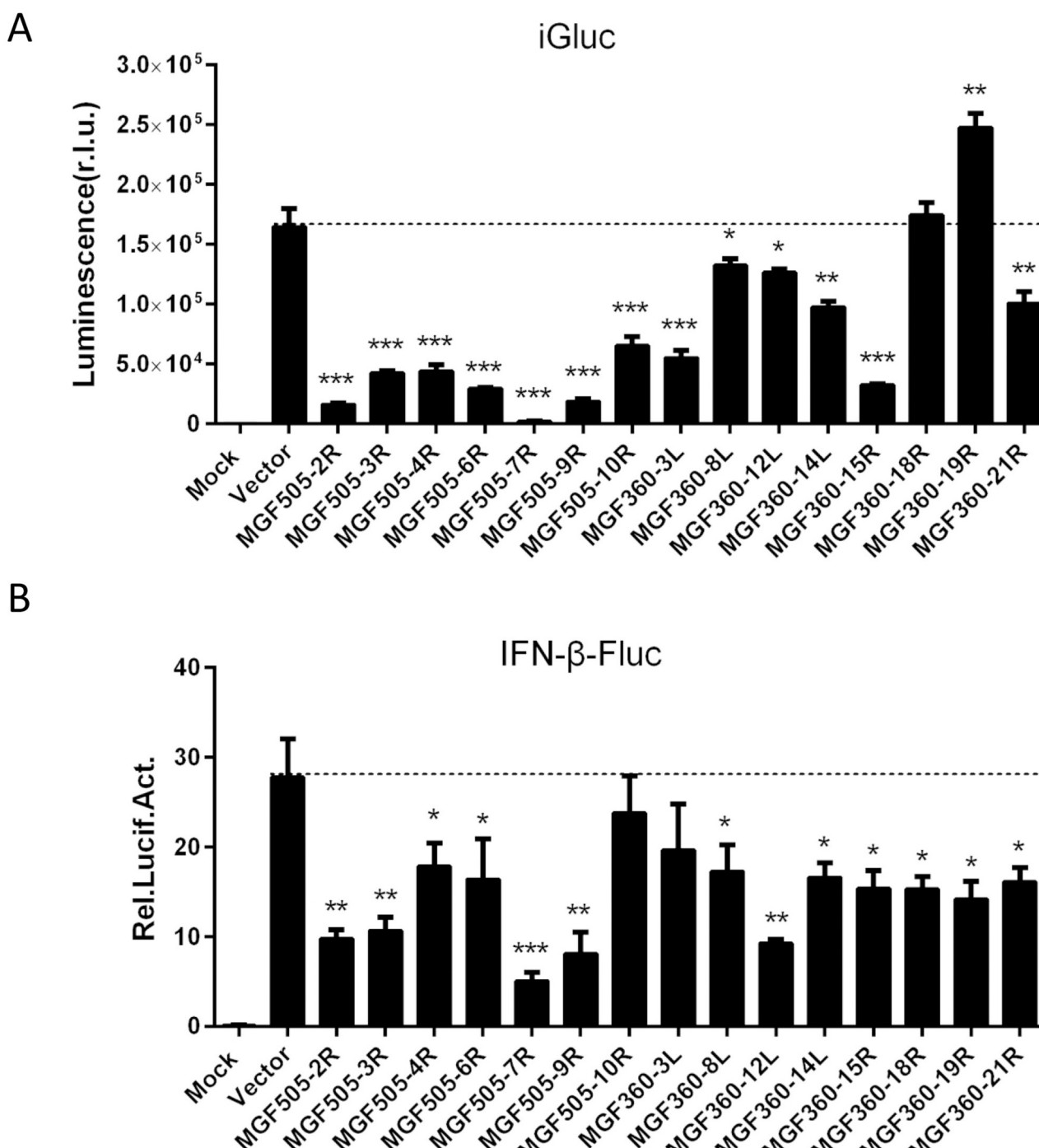

**Fig 3. Screening for ASFV-encoded MGFs that inhibit IL-1β and IFN-β production.** (**A**) HEK293T cells were transfected with a plasmid expressing one of the ASFV-encoded pMGFs in the presence of the iGLuc-based NLRP3 inflammasome system (100 ng iGLuc, 10 ng pCAGGS-caspase-1, 10 ng pCAGGS-ASC, and 12.5 ng pCAGGS-NLRP3), and the supernatants were assessed for luciferase activity at 24 hpt. (**B**) HEK293T cells were transfected with a plasmid expressing one of the ASFV-encoded pMGFs alone with 100 ng of a reporter plasmid and 5 ng of the pRL-TK plasmid for 24 h and then the cells were stimulated with poly(I:C) (1 μg). At 12 hpt, the cells were collected and the luciferase activities were measured. A $p$ value of less than 0.05 was considered statistically significant. $^{*}p<0.05$, $^{**}p<0.01$, $^{***}p<0.001$.

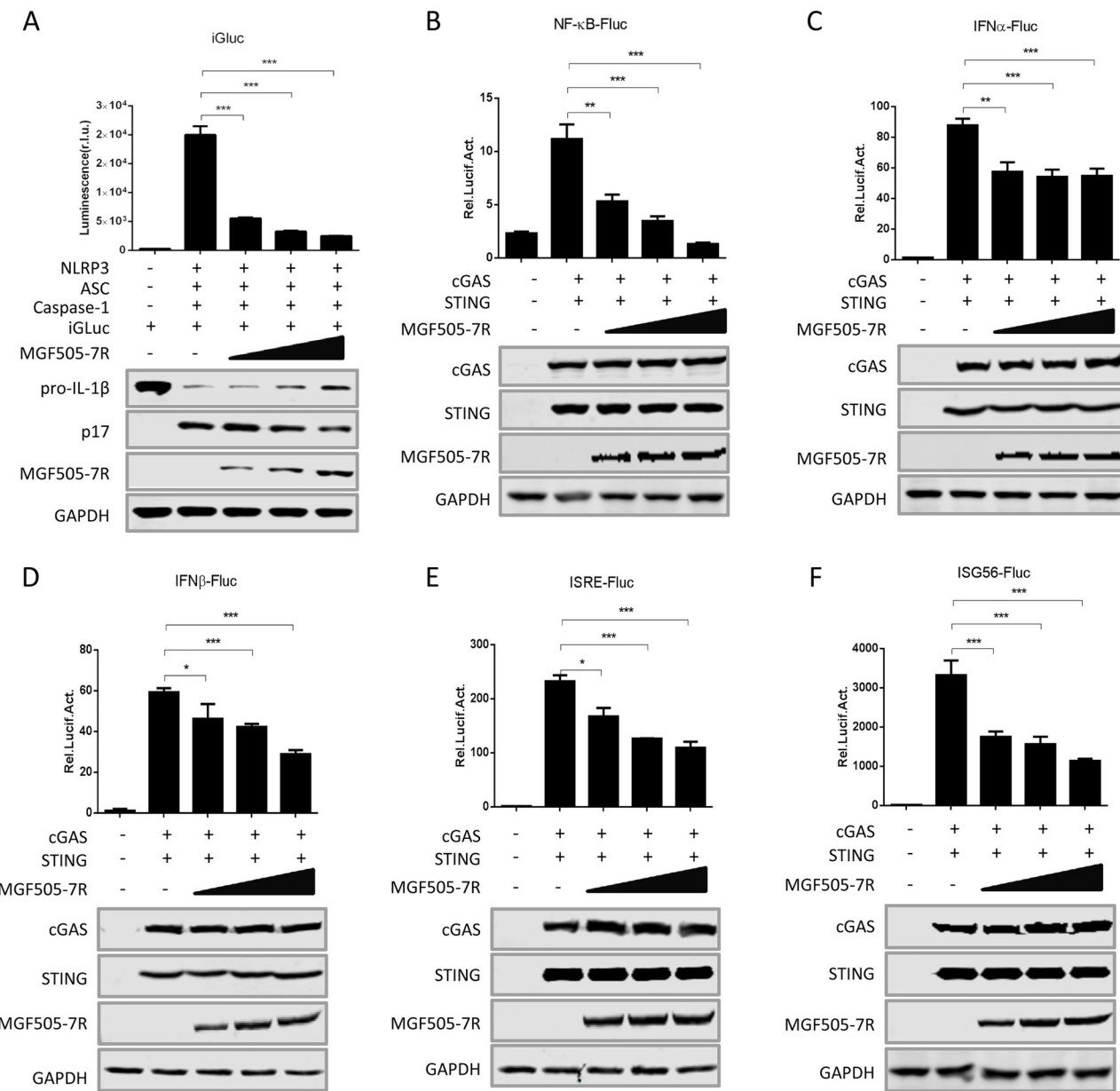

**Fig 4. pMGF505-7R inhibits IL-1β and IFN-β production. (A)** HEK293T cells were transfected with increasing doses of a plasmid expressing pMGF505-7R in the presence of the iGLuc-based NLRP3 inflammasome system, and the supernatants were assessed for luciferase activity at 24 hpt. **(B-F)** HEK293T cells were co-transfected with increasing doses of plasmids expressing pMGF505-7R, along with NF-κB **(B)**, IFN-α **(C)**, IFN-β **(D)**, ISRE **(E)**, or ISG56 **(F)** promoter reporter, and then the cells were stimulated with cGAS/STING for 24 h, the cells were then collected, and the luciferase activities were measured. A *p* value of less than 0.05 was considered statistically significant. $^{*}p<0.05$, $^{**}p<0.01$, $^{***}p<0.001$.

To evaluate the IL-1β and IFN-β levels induced by ASFV-Δ7R, PAMs were infected with ASFV-Δ7R or its parental ASFV-WT at an MOI of 0.5 for 24 or 48 h and the secretion and mRNA expressions of several cytokines were detected by ELISA and qPCR. As we expected, compared with ASFV-WT, ASFV-Δ7R induced much higher levels of IL-1β (Fig 5A). IL-1β secretion induced by ASFV-Δ7R was approximately 8-fold higher than that of ASFV-WT at both time points, and IL-1β mRNA expression was approximately 20-fold higher in PAMs

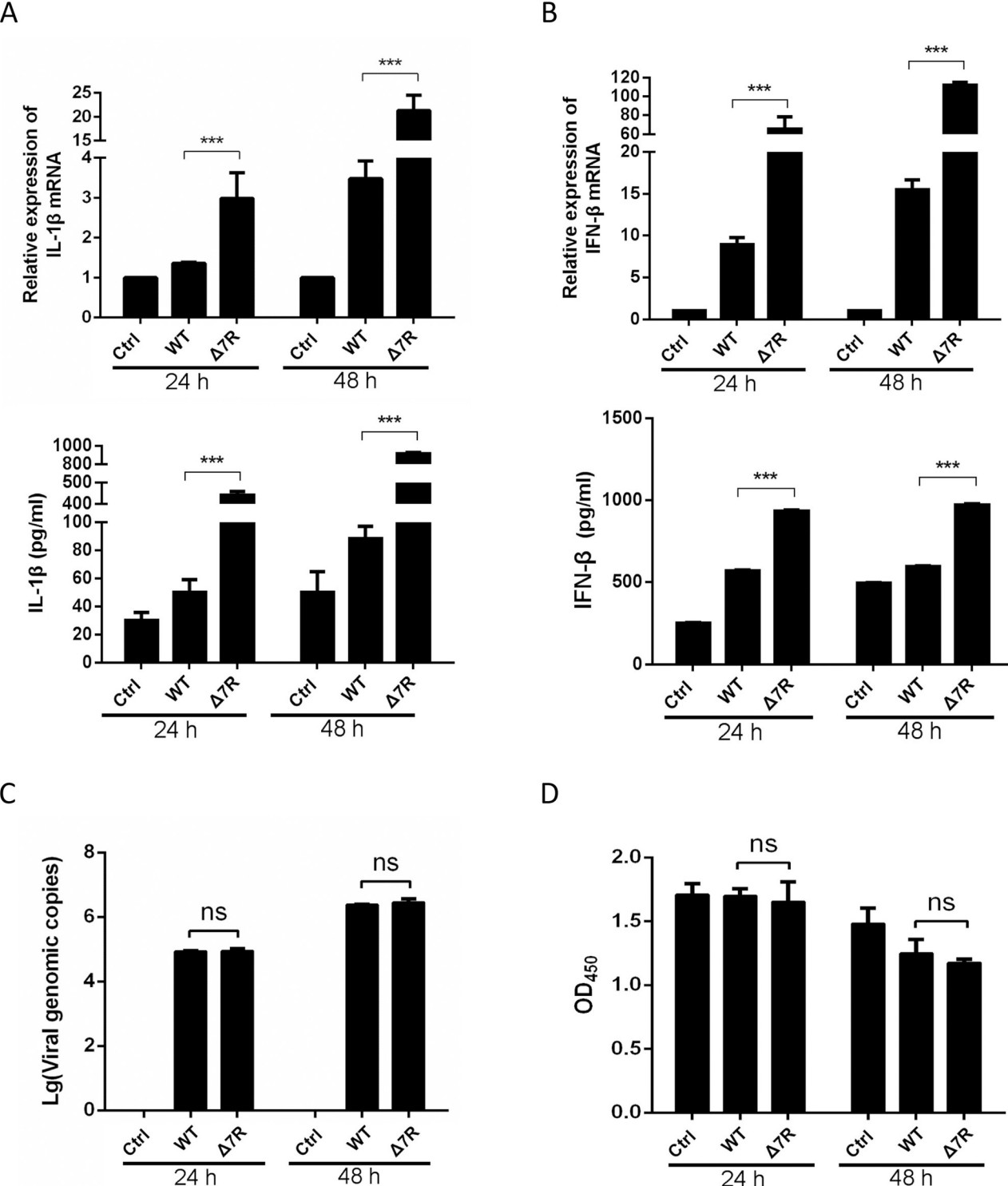

**Fig 5. ASFV-Δ7R induces higher type I IFN and IL-1β production compared with ASFV-WT. (A-D)** PAMs were either mock-infected or infected with ASFV HLJ/18 (ASFV-WT) or ASFV-Δ7R at an MOI of 0.5. At 24 or 48 hpi, the IL-1β **(A)** and IFN-β **(B)** levels in the cell culture supernatants were detected by ELISA and the mRNA levels in the cell lysates were determined by qPCR. The genomic copy numbers of ASFV in PAMs were measured by qPCR **(C)** and the cell viabilities were analyzed using a CCK-8 counting kit **(D)**. A $p$ value of less than 0.05 was considered statistically significant. $^{*}p<0.05$, $^{**}p<0.01$, $^{***}p<0.001$.

infected with ASFV-Δ7R at 48 hpi (Fig 5A). Similarly, we found that ASFV-Δ7R induced much higher levels of IFN-β secretion and mRNA expression compared with ASFV-WT (Fig 5B). To exclude the possibility that the enhanced inductions of IL-1β and IFN-β by ASFV-Δ7R was due to more virus particles or increased host cell viability instead of loss of functions of pMGF505-7R, we detected the viral genomic copy numbers and cellular viabilities following viral infections. The results showed that the genomic copy number of ASFV-Δ7R was at a similar level compared to that of its parental ASFV-WT (Fig 5C), and the cellular viabilities of PAMs had no obvious difference following the infections of ASFV-Δ7R and ASFV-WT (Fig 5D). Taken together, our findings reveal that pMGF505-7R plays pivotal roles to inhibit IL-1β and type I IFN production in PAMs upon ASFV infection.

## ASFV pMGF505-7R interacts with IKK complex to inhibit IL-1β transcription

During viral infection and replication, the "first signaling" for inflammasome activity comes from the activation of PRRs by pathogen-associated molecular patterns (PAMPs), leading to transcription of pro-IL-1β [27]. Toll-like receptors (TLRs) are important PRRs involved in pro-IL-1β transcription, and at least 9 members of the porcine TLR family have been identified. In order to investigate the role of TLR signaling pathway in ASFV infection-induced pro-IL-1β transcription, specific siRNAs targeting each of these TLRs and MyD88 were synthesized. PAMs were transfected with specific siRNAs or non-targeting control siRNA, followed by ASFV-Δ7R infection. As shown in S4A Fig, knockdown of TLR1, TLR2, TLR3, TLR4, TLR9 and MyD88 significantly decreased ASFV-Δ7R-induced pro-IL-1β transcription. The knockdown efficiencies of siRNAs against these receptors were confirmed by qPCR (S4B Fig). Combined with previous results shown that pMGF505-7R strongly inhibited NF-κB activation (Fig 4B) and ASFV-Δ7R infection induced much higher levels of IL-1β mRNA transcription compared with ASFV-WT (Fig 5A), our results suggest that ASFV-Δ7R induces IL-1β production through TLRs signaling pathway and pMGF505-7R functions to block the "first signaling" of inflammatory responses.

To investigate the molecular mechanisms underlying pMGF505-7R inhibition of NF-κB activation, the effects of pMGF505-7R on NF-κB promoter activation mediated by LPS and various adaptors were examined. The results showed that ectopically expressed pMGF505-7R strongly inhibited NF-κB reporter activation induced by LPS, IKKα and IKKβ in a dose-dependent manner (Fig 6A–6C). These results indicate that pMGF505-7R might negatively regulate NF-κB reporter activation by targeting IKK complex. To evaluate whether pMGF505-7R interacts with IKK complex, the immunoprecipitation assay was performed in HEK293T cells to evaluate the interaction between pMGF505-7R and each of the core components of the IKK complex, namely IKKα, IKKβ and NEMO. As shown in Fig 6D, IKKα, but not IKKβ and NEMO, co-precipitated with pMGF505-7R.

To further evaluate the interaction between endogenous IKKα and MGF505-7R in the condition of ASFV infection, a genetically modified ASFV with N-terminally EGFP-tagged MGF505-7R (ASFV-EGFP-7R) was generated (S5A–S5C Fig). EGFP-MGF505-7R fusion gene expression was detected in ASFV-EGFP-7R-infected PAMs (S5D Fig and S2 Table) and ASF-V-EGFP-7R displayed a similar growth kinetic compared to its parental ASFV-WT (S5E Fig). To rule out the disruption of EGFP insertion on the function of pMGF505-7R, we constructed a plasmid expressing EGFP-MGF505-7R and transfected the HEK293T cells with the plasmid in combination with iGLuc-based inflammation reconstruction system or IFN-β reporter. The results showed that EGFP-MGF505-7R inhibited both IL-1β and IFN-β production in a similar manner compared with pMGF505-7R (S5F and S5G Fig), suggesting that EGFP insertion did

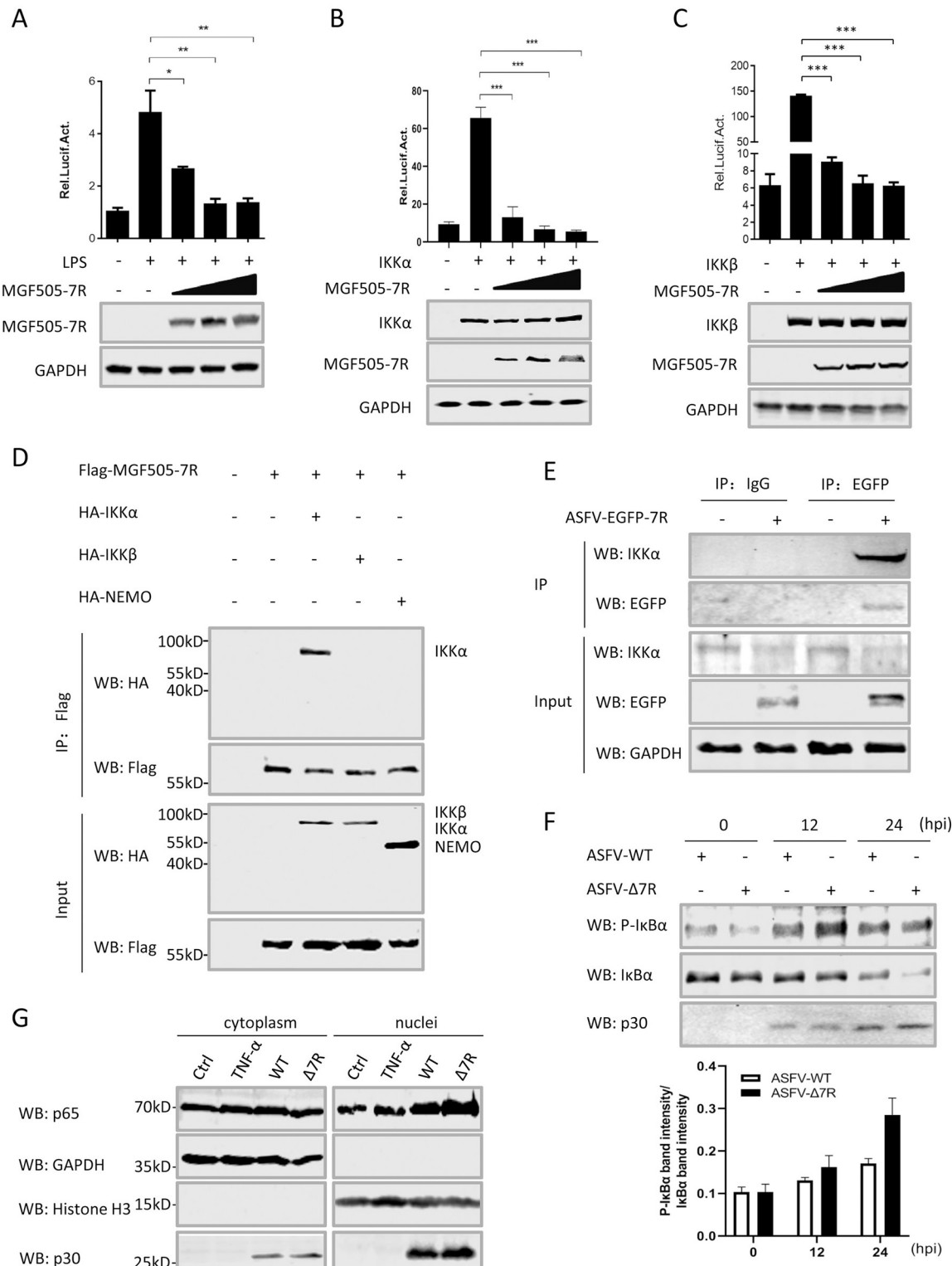

**Fig 6. ASFV pMGF505-7R interacts with IKK complex to inhibit IL-1β transcription. (A–C)** HEK293T cells were transfected with NF-κB reporter, increasing amounts of a plasmid encoding pMGF505-7R along with stimulation of LPS (100 ng/mL) **(A)** or a plasmid encoding IKKα **(B)** or IKKβ **(C)**. The luciferase activities were measured. The expressions of IKKα, IKKβ and pMGF505-7R were detected by Western blotting. GAPDH expressions were detected as a loading control. **(D)** HEK293T cells were transfected with a plasmid encoding Flag-tagged pMGF505-7R along with a plasmid encoding HA-tagged IKKα, IKKβ or NEMO as indicated. 36 hpt, the

cells were lysed and whole cell lysates (WCL) were immunoprecipitated with anti-Flag mAb. The immunoprecipitants were detected by Western blotting with antibodies indicated. **(E)** PAMs were either mock-infected or infected with ASFV-EGFP-7R (1 MOI) for 36 h, and then a Co-IP was performed with anti-EGFP antibody. Immunoglobulin G (IgG) was used as a negative control. **(F)** PAMs were infected with ASFV-WT (1 MOI) or ASFV-Δ7R (1 MOI) for 12 or 24 h, and the phosphorylation levels of IκBα were analyzed by Western blot. The relative protein band intensities analyzed by Image studio. **(G)** PAMs were treated with TNF-α (15 ng/mL) or infected with ASFV-WT (WT) or ASFV-Δ7R (Δ7R), and p65 in the nuclear and cytoplasmic compartments, and ASFV-p30 were detected by Western blotting. Histone H3 and GAPDH were used as nuclear and cytosolic markers. A $p$ value of less than 0.05 was considered statistically significant. $^{*}p<0.05$, $^{**}p<0.01$, $^{***}p<0.001$.

not affect the function of pMGF505-7R. Then we used the ASFV-EGFP-7R to infect PAMs and performed the immunoprecipitation assay to evaluate whether pMGF505-7R interacts with IKKα in the context of ASFV-EGFP-7R infection in PAMs. As shown in Fig 6E, endogenous IKKα co-precipitated with EGFP-MGF505-7R in the ASFV-EGFP-7R-infected PAMs.

Previous studies showed that the degradation of the phosphorylated IκBα is required for the activation of IKKα and IKKβ [28]. Therefore, we tested whether pMGF505-7R had a role in inhibiting the phosphorylation of IκBα. PAMs were infected with ASFV-WT or ASFV-Δ7R, and the phosphorylation levels of IκBα at 12 and 24 hpi were analyzed. The results showed the phosphorylation levels of IκBα in ASFV-Δ7R-infected PAMs were higher than that of ASFV-WT at different time points, suggesting pMGF505-7R inhibits IκBα phosphorylation during ASFV infection (Fig 6F). Because NF-κB activation leads to p65 nuclear translocation, therefore we tested the effect of pMGF505-7R on nuclear translocation of p65 by Western blotting. As shown in Fig 6G, the amount of nuclear-translocated p65 was obviously increased following ASFV-Δ7R infection. Taken together, our results demonstrate that ASFV pMGF505-7R inhibits NF-κB activation through interacting with IKKα, leading to reduce IL-1β production.

## ASFV pMGF505-7R targets NLRP3 to inhibit IL-1β maturation

The "second signaling" for IL-1β production results from the formation of inflammasome and active caspase-1. It has been reported that DNA virus activate NLRP3 and AIM2 inflammasomes to induce IL-1β production in mice [29,30]. Although pig does not have AIM2 and AIM2-like proteins, PAMs still sense intracellular dsDNA [31]. Therefore, we proposed that ASFV infection might induce IL-1β production through activation of NLRP3 inflammasome. To test our hypothesis, PAMs were transfected with specific siRNAs targeting NLRP3, and then the cells were infected with ASFV-Δ7R. The knockdown efficiencies of siRNAs against NLRP3 were confirmed by qPCR and Western blotting. As shown in S6A and S6B Fig, IL-1β secretion induced by ASFV-Δ7R was significantly decreased in the NLRP3-knockdown PAMs, indicating that NLRP3 plays essential role in ASFV-induced IL-1β activation.

To further investigate whether pMGF505-7R inhibits NLRP3 inflammasome formation, the NLRP3 inflammasome reconstruction system based on iGluc reporter and a plasmid expressing pMGF505-7R were transfected into HEK293T cells, and then the IL-1β levels were detected by measuring luciferase activity. The results showed that pMGF505-7R significantly inhibited iGluc reporter activity, especially in the presence of all inflammasome components (Fig 7A). To test whether pMGF505-7R physically interacts with the components of NLRP3 inflammasome, Co-IP assay evaluating the interaction between pMGF505-7R and NLRP3, ASC or caspase-1 was performed. As shown in Fig 7B, pMGF505-7R co-precipitated with NLRP3, but not ASC or caspase-1. Consistent with these results, NLRP3 and pMGF505-7R were found to be both localized in the cytoplasm of HEK293T cells (S7A Fig). To map which domain of NLRP3 is required for the interaction between NLRP3 and pMGF505-7R, four NLRP3 truncated mutants were constructed, and Co-IP assay was repeated. The result showed the

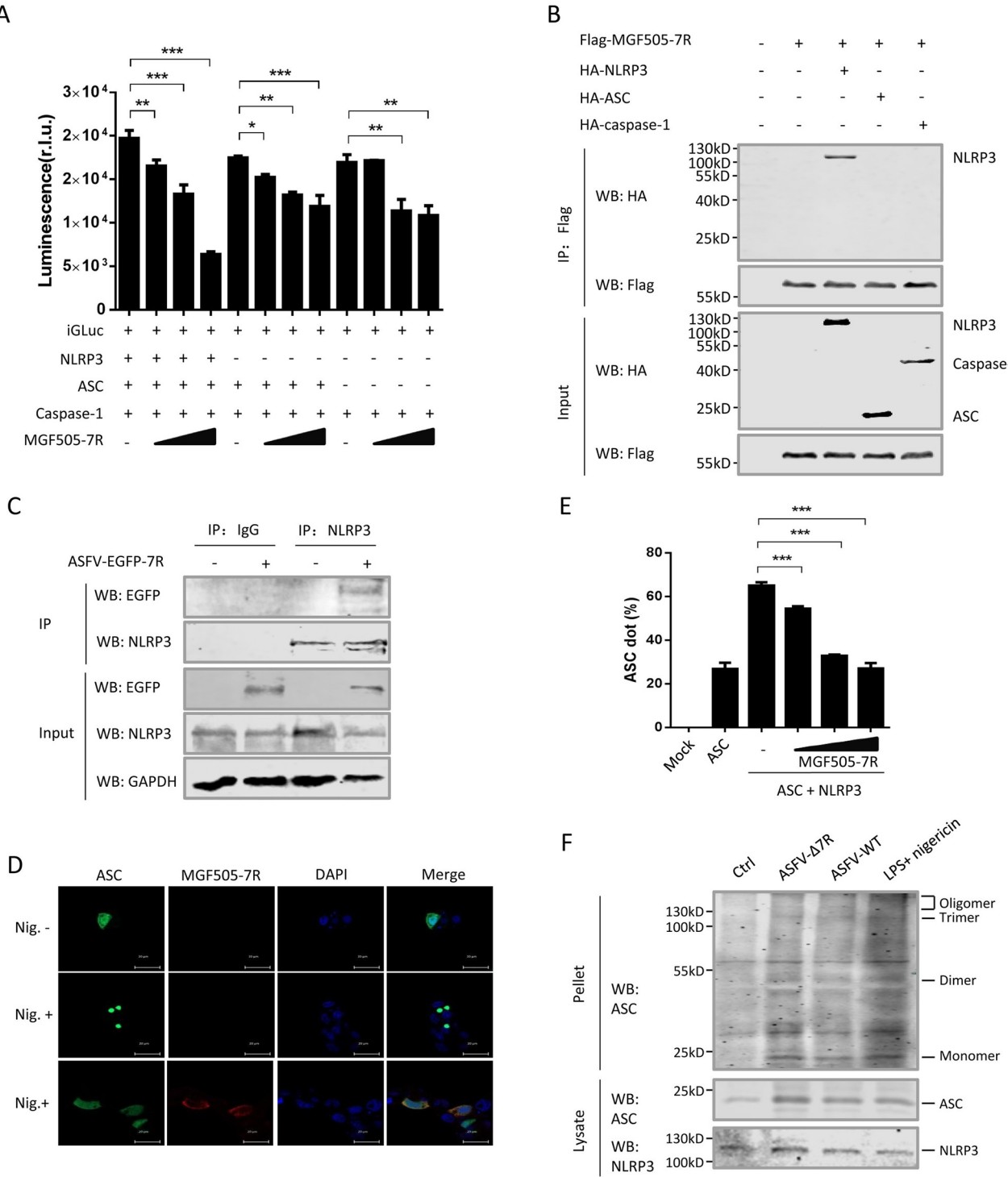

**Fig 7. pMGF505-7R targets NLRP3 to inhibit inflammasome complex formation. (A)** HEK293T cells were transfected with different doses of MGF505-7R in the presence of the iGLuc-based NLRP3 inflammasome system or the system lacking NLRP3 or both NLRP3 and ASC, and the supernatants were assessed for luciferase activity at 24 h after transfection. **(B)** HEK293T cells were transfected with a plasmid encoding pMGF505-7R along with a plasmid encoding HA-tagged NLRP3, ASC or caspase-1. 36 hpt, the cells were lysed and whole cell lysates (WCL) were immunoprecipitated with anti-Flag mAb. The immunoprecipitants were detected by immunoblotting with antibodies indicated. **(C)** PAMs were either mock-infected or infected with ASFV-EGFP-7R (1 MOI) for 36 h, and then a Co-IP was performed with anti-NLRP3 antibody. IgG was used as a negative control. **(D)** The PAM cell line 3D4/21 (CRL-2843) were transfected with a plasmid encoding GFP-ASC alone or together with Flag-MGF505-7R. At 24 hpt, cells were stimulated with Nigericin (5 μM) for another 4 h. The cells were then fixed and probed with rabbit anti-Flag mAb and GFP,

and nucleus marker DAPI, and then observed by confocal microscopy. **(E)** HEK293T cells were transfected with GFP-ASC, HA-NLRP3 or Flag-MGF505-7R. 24 hpt, the cells were harvested, fixed, and the fraction of cells containing ASC specks was quantified by flow cytometry. Average values of cells with ASC specks came from three independent experiments. **(F)** PAMs were infected with ASFV-Δ7R or ASFV-WT. The cell lysates were prepared and the pellets were washed with PBS for three times and cross-linked using fresh DSS for Western blotting. A *p* value of less than 0.05 was considered statistically significant. *$p<0.05$, **$p<0.01$, ***$p<0.001$.

NACHT and LRR domains were involved in the interaction between NLRP3 and pMGF505-7R (S7B Fig). To further evaluate the interaction between pMGF505-7R and NLPR3 in the context of ASFV infection, PAMs were infected with ASFV-EGFP-7R and the cell lysates were coprecipitated with anti-NLRP3 antibody. As shown in Fig 7C, EGFP-MGF505-7R was pulled down by endogenous NLRP3 in the ASFV-infected PAMs. These results suggest that NLRP3 is physically targeted by pMGF505-7R. To test whether pMGF505-7R can inhibit NLRP3 inflammasome assembly, GFP-ASC was used as a reporter for inflammasome assembly in 3D4/21 cells. Without stimulation of Nigericin, a bacterial toxin used widely to activate NLPR3 inflammasome, GFP-ASC alone was expressed ubiquitously in the cell without appearance of any speck-like aggregates. Nigericin treatment induced characteristic ASC specks, indicating assembly of NLRP3 inflammasome. Interestingly, co-expression of pMGF505-7R with GFP-ASC totally inhibited ASC specks formation in the presence of Nigericin treatment, indicating that pMGF505-7R inhibits NLRP3 inflammasome assembly (Fig 7D). ASC speck formation can be detected by the characteristic redistribution of GFP-ASC fluorescence measured by flow cytometry [32]. We founded that overexpression of NLRP3 led to substantial ASC speck formation, which was significantly inhibited by pMGF505-7R in a dose dependent manner (Fig 7E and S7C Fig). Furthermore, we noticed that ASFV-Δ7R infection induced higher levels of ASC oligomer than that of ASFV-WT infection in PAMs (Fig 7F). These findings demonstrate that pMGF505-7R interacts with NLRP3 to inhibit NLRP3 inflammasome assembly, leading to decreased IL-1β production.

## pMGF505-7R targets IRF3 to inhibit IFN production

We have shown that pMGF505-7R inhibited type I IFN production (Fig 4C–4F). To investigate how pMGF505-7R exerts its inhibition, pMGF505-7R was co-expressed with different adaptors in type I IFN signaling and then the IFN-β promoter activities were examined. We found that the ectopically expressed pMGF505-7R strongly inhibited IFN-β reporter activation induced by cGAS/STING (Fig 4D), TBK1 (Fig 8A), IRF3 (Fig 8B), and IRF3-5D (Fig 8C), a constitutively active form of IRF3, in a dose-dependent manner. These results indicated that pMGF505-7R might inhibit type I IFN production by targeting IRF3. To explore whether pMGF505-7R interacts with IRF3 or IRF3-5D, Co-IP assay between pMGF505-7R and IRF3 or IRF3-5D was performed. We found that pMGF505-7R precipitated with both IRF3 (Fig 8D) and IRF3-5D (Fig 8E). To further evaluate the interaction between pMGF505-7R and IRF3 in the context of ASFV infection, PAMs were infected with ASFV-EGFP-7R and the cell lysates were immunoprecipitated with anti-IRF3 antibody. The result showed that EGFP-MGF505-7R interacted with endogenous IRF3 in ASFV-infected PAMs (Fig 8F), indicating IRF3 may be targeted by pMGF505-7R for suppressing type I IFN signaling. Then, we tested whether pMGF505-7R affects IRF3 nuclear translocation, a critical step in the signaling for inducing type I IFN production. Under normal condition, the IRF3 mainly distributes in the cytoplasm. With the stimulation of SeV, a large portion of IRF3 translocated to the nucleus, which was dramatically blocked by pMGF505-7R (Fig 8G), indicating functional suppression of IRF3 by pMGF505-7R. Meanwhile, the effect of pMGF505-7R on nuclear translocation of IRF3 was analyzed by Western blotting. PAMs were infected with SeV, ASFV-WT or ASFV-

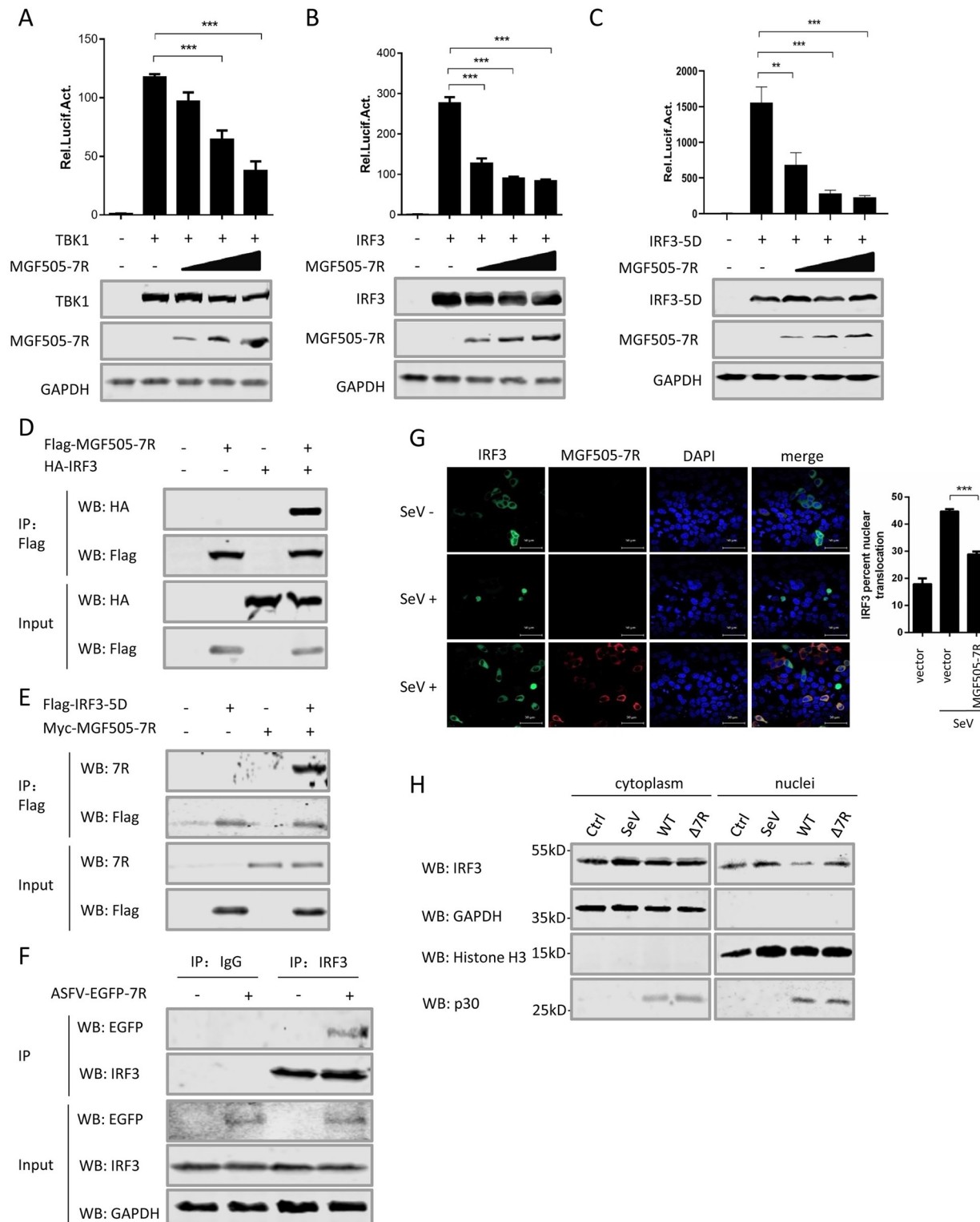

**Fig 8. pMGF505-7R targets IRF3 to inhibit IFN-β production. (A-C)** HEK293T cells were transfected with IFN-β reporter, different amounts (0, 100, 200 or 400 ng) of a plasmid encoding pMGF505-7R, together with a plasmid encoding TBK1 **(A)**, IRF3 **(B)** or IRF3-5D **(C)**. 24 hpt, the cells were lysed and the activities of the IFN-β reporter were assessed by the luciferase assay. The expressions of TBK1, IRF3 and pMGF505-7R were detected by Western blotting. GAPDH served as a loading control. **(D-E)** HEK293T cells were transfected with a plasmid encoding pMGF505-7R along with a plasmid encoding HA-IRF3 or HA-IRF3-5D as indicated. 36 hpt, the cells were lysed and whole cell lysates were immunoprecipitated

with anti-Flag mAb. The immunoprecipitants were detected by immunoblotting with antibodies indicated. **(F)** PAMs were either mock-infected or infected with ASFV-EGFP-7R for 36 h, and then a Co-IP was performed with anti-IRF3 antibody. IgG was used as a negative control. **(G)** HeLa cells were transfected with GFP-IRF3 and Flag-MGF505-7R. At 24 hpt, the cells were stimulated with SeV for another 12 h and then probed with rabbit anti-Flag mAb and GFP, stained with nucleus marker DAPI and subsequently observed by confocal microscopy. The percentages of IRF3 nuclear translocation were quantified. **(H)** PAMs were infected with SeV, ASFV-WT (WT) or ASFV-Δ7R (Δ7R), and IRF3 in the nuclear and cytoplasmic compartments and ASFV-p30 were detected by Western blotting. Histone H3 and GAPDH were used as nuclear and cytosolic markers. A *p* value of less than 0.05 was considered statistically significant. $^{*}p<0.05$, $^{**}p<0.01$, $^{***}p<0.001$.

Δ7R, and then the nuclear and cytoplasmic extracts of the cells were harvested and subjected to IRF3 analysis. As shown in Fig 8H, the amount of nuclear-translocated IRF3 was obviously increased following ASFV-Δ7R infection compared with that of ASFV-WT infection. Taken together, these results demonstrate that pMGF505-7R targets IRF3 to block type I IFN production.

## Deletion of MGF505-7R attenuates ASFV virulence in pigs

To further confirm the virulence of ASFV-Δ7R and thus the biological functions of pMGF505-7R *in vivo*, pigs were challenged intramuscularly (i.m.) with ASFV-Δ7R or ASFV-WT at a dose of $10^3$ or $10^5$ $HAD_{50}$, the symptoms, survivals and pathological changes were recorded and assessed. Following viral challenge, the body temperatures of all animals increased gradually, peaked at 41.7°C by day 9 and then gradually decreased until day 19 post infection compared with control group (Fig 9A). Pigs inoculated with ASFV-WT began to die from day 8 after virus inoculation, and all died on day 10 (Fig 9B). However, the pigs inoculated with ASFV-Δ7R started to die from the day 12 or 13 and over 60% of them eventually survived from the challenge. In order to study the distribution of the virus, pigs survived at day 19 post inoculation were euthanized, and tissues including the heart, lung, spleen, tonsil, thymus and five lymph nodes (intestinal lymph node, inguinal lymph node, submaxillary lymph node, bronchial lymph node, and gastrohepatic lymph node) were collected from all pigs for viral DNA quantification by qPCR. The results showed that pigs from the $10^3$ and $10^5$ $HAD_{50}$ ASFV-Δ7R-inoculated groups had much lower copies of viral DNA compared with the ASFV-WT group (Fig 9C). It is noteworthy that the viral DNA copies in the $10^3$ $HAD_{50}$ ASFV-Δ7R-inoculated group decreased 2 to 3 orders of magnitude (Fig 9C). These results indicate that the virulence of ASFV-Δ7R is effectively reduced, proving a critical role of the MGF505-7R gene in determining the virulence of ASFV.

## ASFV-Δ7R induces higher type I IFN and IL-1β production in pigs

A previous study reported that pigs infected with ASFV genotype II have increased proinflammatory cytokines in their sera [14]. Our findings have revealed that the attenuated recombinant ASFV-Δ7R induced high levels of IL-1β and type I IFN production in PAMs (Fig 5A and 5B). Therefore, we attempted to explore whether ASFV-Δ7R challenge enhances IL-1β and type I IFN production in pigs. Randomly assigned groups of pigs were mock challenged with PBS or challenged intramuscularly (i.m.) with ASFV-Δ7R ($10^3$ and $10^5$ $HAD_{50}$) or ASFV-WT ($10^3$ $HAD_{50}$), and the serum samples were collected at day 1, day 5 and day 8 post challenge for evaluation of the levels of IL-1β, IFN-α and IFN-β. For induction levels of IFN-β, there was no obvious difference between the ASFV-WT challenge group and ASFV-Δ7R challenge group at day 1 and day 5; however, at day 8 the IFN-β levels in the $10^3$ $HAD_{50}$ ASFV-Δ7R challenge group was significantly increased (Fig 10A). For IFN-α, ASFV-Δ7R challenge groups had significantly higher levels at day 5 (Fig 10B), and IL-1β was also obviously increased in ASFV-Δ7R challenge groups at day 5 and day 8 (Fig 10C). These results demonstrated that the attenuated ASFV-Δ7R induces higher IL-1β and type I IFN production in pigs compared with

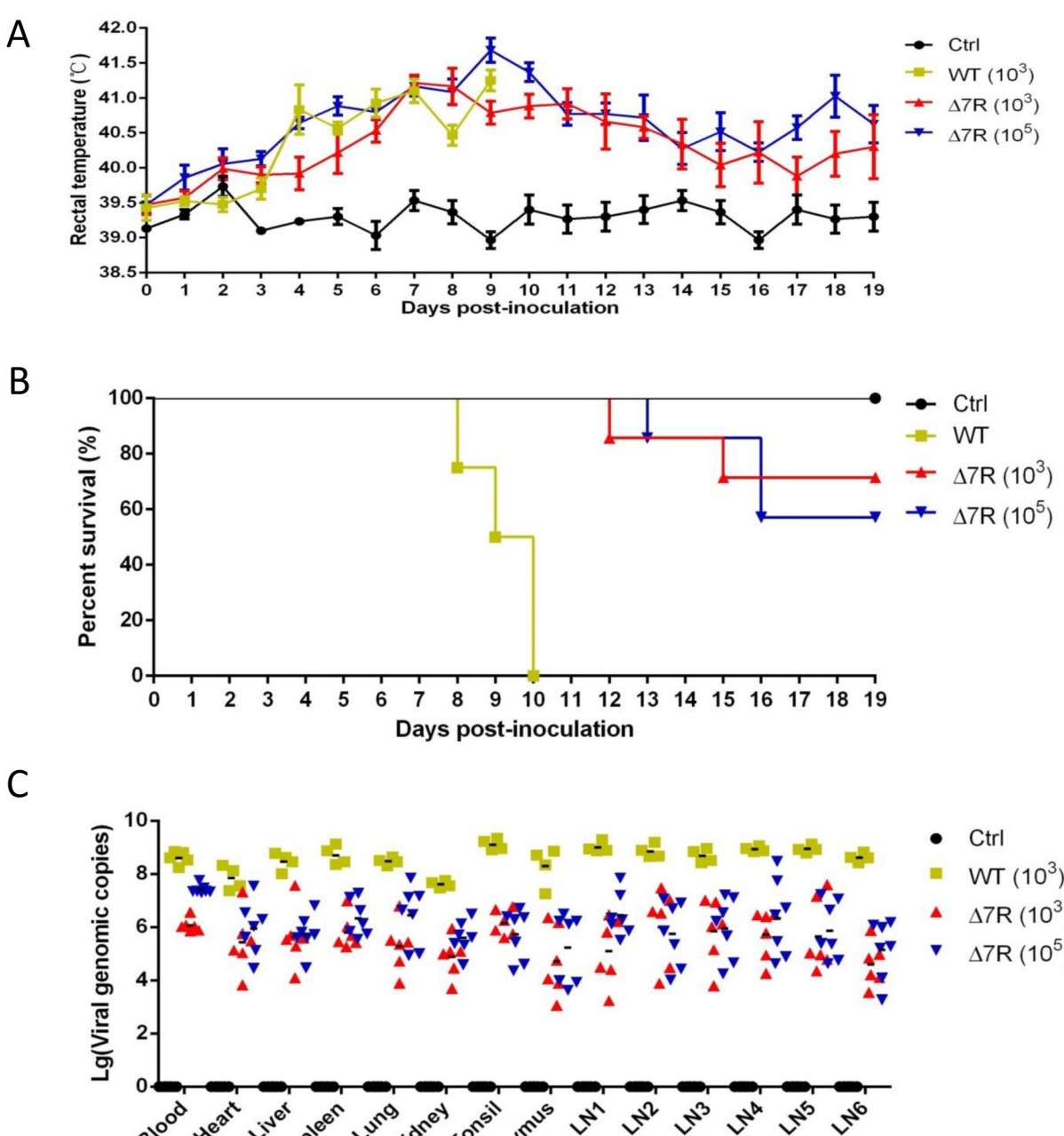

**Fig 9. Deletion of MGF505-7R attenuates ASFV virulence in pigs. (A-C)** The rectal temperature measurements (**A**) and survival rates (**B**) for the different groups of pigs unchallenged (Ctrl, n = 4), or challenged with $10^3$ HAD$_{50}$ of parental ASFV-WT (WT, n = 4), $10^3$ HAD$_{50}$ of ASFV-Δ7R (Δ7R, n = 7) or $10^5$ HAD$_{50}$ of ASFV-Δ7R (Δ7R, n = 7) were recorded daily till day 19 post challenge. At day 19, the surviving pigs were euthanized, and the blood and tissues including the heart, lung, spleen, tonsil, thymus and five lymph nodes (intestinal lymph node, inguinal lymph node, submaxillary lymph node, bronchial lymph node, and gastrohepatic lymph node) were collected from all pigs for viral DNA quantification by qPCR (**C**).

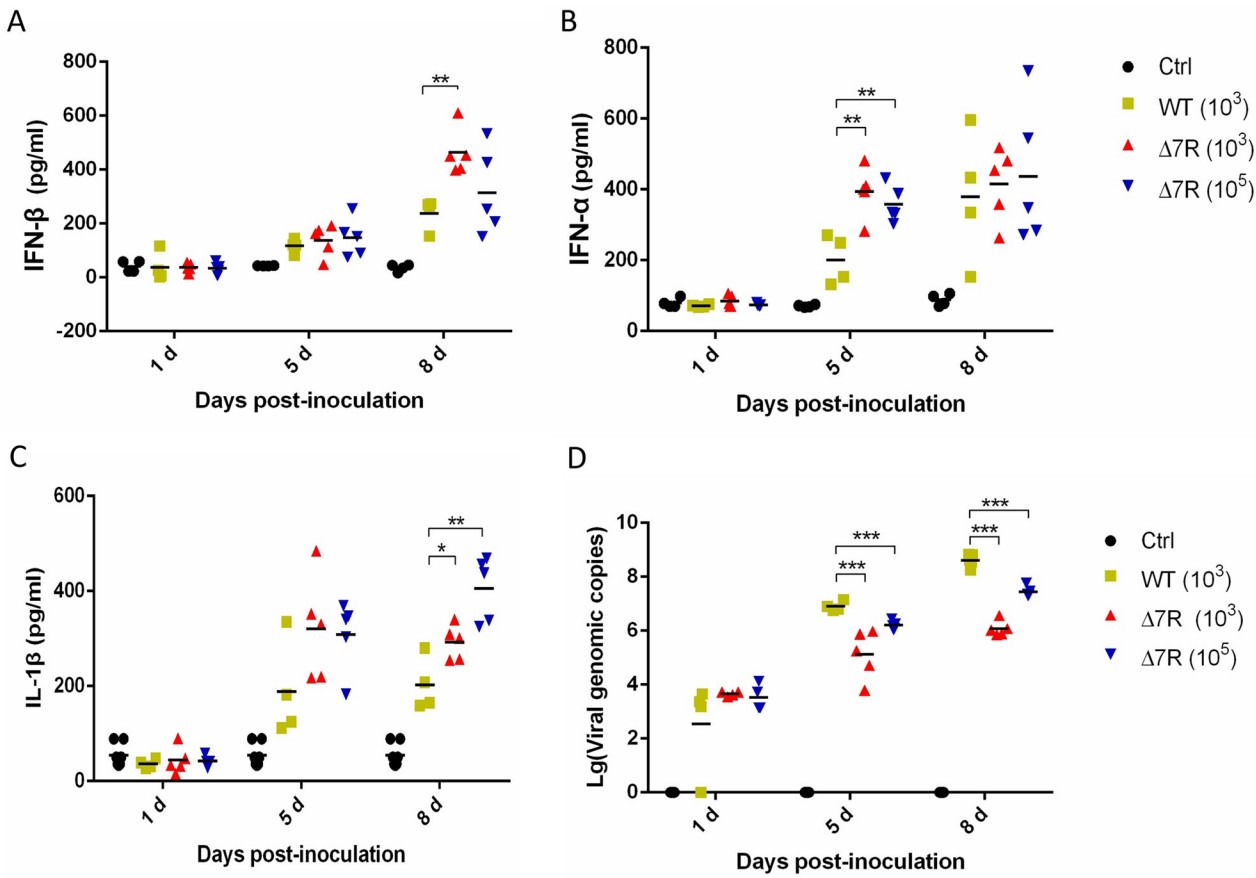

**Fig 10. ASFV-Δ7R infection induces higher type I IFN and IL-1β production in pigs. (A-D)** Pigs were mock challenged with PBS (Ctrl, n = 4) or challenged intramuscularly with ASFV-Δ7R (Δ7R, n = 5, $10^3$ and $10^5$ $HAD_{50}$) or parental ASFV-WT (WT, n = 4, $10^3$ $HAD_{50}$), and the sera samples were collected at day 1, day 5 and day 8 post challenge. The protein levels of IFN-β **(A)**, IFN-α **(B)** and IL-1β **(C)** in the serum samples were measured with commercial ELISA kits. The virus loads in the serum samples collected at day 1, day 5 and day 8 post challenge. were evaluated using qPCR. **(D)** Data presented are mean ± SD. **, $P < 0.01$, *, $P < 0.05$, student's *t-test*, in comparison with control groups during statistical analysis.

ASFV-WT. We then evaluated the virus loads in the serum samples using qPCR, and found that the viral DNA copies in the ASFV-Δ7R challenge groups were lower than those in the ASFV-WT group at both day 5 and day 8 post challenge (Fig 10D). Taken together, these results indicate that the attenuated ASFV-Δ7R induces higher type I IFN and IL-1β production in pigs, which reduced the viral load and its pathogenicity.

## Discussion

The innate immune system is a highly conserved signaling network that is important for clearing the invading pathogens to protect the host [33]. During the process of viral infection, PRRs in the host cells recognize conserved microbial components called PAMPs and trigger a series of signals that induce production of inflammatory cytokine IL-1β, type I IFNs and other downstream effectors to antagonize viral infection [33]. IL-1β production requires two separate signals: signal 1 is involved in priming of the host cell through detection of PAMPs by its PRRs, leading to increase of the pro-IL-1β transcription while signal 2 is related to the activation of several inflammasomes, resulting in cleavage of pro-IL-1β into mature IL-1β. In the study, we proved that ASFV infection induced IL-1β production both in PAMs and in pigs (Figs 1 and 10),

although it is lower level compared with HSV-1 infection. A large number of viruses have been reported to induce the production of IL-1β during infection [29]. For examples, porcine reproductive and respiratory syndrome virus (PRRSV) induces pro-IL-1β transcription through the TLR4-mediated MyD88 signaling pathway and promotes IL-1β secretion through the NLRP3 inflammasome [34]. Vaccinia virus and murine cytomegalovirus infections stimulate the formation of the absent in melanoma 2 (AIM2) inflammasome to activate caspase-1-dependent maturation of IL-1β [35]. Pigs do not have the AIM2 gene, but they still can respond to intracellular dsDNA stimulation and produce IL-1β, suggesting presence of a cytoplasmic DNA sensor in pigs to directly activate the inflammatory response [31]. It has been reported that NLRP3 is involved in DNA-induced inflammation [30]. To investigate how ASFV induces inflammasome activation, we used specific siRNAs targeting NLRP3, MyD88 and different TLRs and found that ASFV infection induces IL-1β production through TLRs/MyD88 pathway and NLRP3 inflammasome in PAMs (S4 and S6 Figs). These results reveal ASFV infection induces NLRP3 inflammasome activity to release inflammatory factors in PAMs.

Inflammatory responses counter viral replication and remove infected immune cells through an inflammatory cell death program termed as pyroptosis. As a countermeasure, viruses have evolved to antagonize host inflammasome pathways. Several studies have revealed different mechanisms utilized by viruses to suppress the inflammasome activation, which may occur at different steps in the replication cycles of the viruses. One such mechanisms is involved in inhibition of the translocation and activation of NF-κB, which limits the synthesis of NLRP3 and the inflammasome substrates, such as IL-1β and IL-18. For example, human cytomegalovirus (HCMV)-encoded protein pp65 inhibits the transcriptional activity of NF-κB to downregulate IL-1β, and mediates HCMV immune evasion through regulation of inflammasome activation [36]. ORF3 of hepatitis E virus (HEV) inhibits the expression of proinflammatory cytokines and chemotactic factors in LPS-stimulated human PMA-THP1 cells by inhibiting NF-κB pathway [37]. Another mechanism is commonly used to inhibit the inflammasome assembly through the multifunctional viral-encoded proteins. For examples, SeV V protein inhibits the assembly of NLRP3 inflammasome, including inhibition of the NLRP3-dependent ASC oligomerization, NLRP3 self-oligomerization, and intermolecular interactions between NLRP3 molecules [38]. The leporipoxvirus encodes the PYD-containing protein M13L-PYD, which interacts with and inhibits ASC, thereby suppressing caspase-1-dependent IL-1β production [39]. Poxviruses and herpesviruses have large DNA genomes that encode various protein products, several of them inhibit the assembly of the inflammasome by preventing its oligomerization [40]. However, the study on the regulation of inflammatory responses by ASFV is not fully understood. In this study, we found that ASFV uses both mechanisms to antagonize IL-1β production. ASFV is a large cytoplasmic DNA virus whose genome encodes up to 150 proteins, and most members of pMGFs we tested can inhibit IL-1β production (Fig 3A). Among them, pMGF505-7R has the strongest inhibitory effect on IL-1β production (Fig 3A). pMGF505-7R interacts with IKKα to inhibit NF-κB activation, leading to reduced pro-IL-1β transcription and production (Figs 4 and 6); meanwhile, pMGF505-7R binds to NLRP3 to inhibit NLRP3 inflammasome assembly, leading to decreased IL-1β maturation and secretion (Fig 7). Consistent with these results, ASFV-Δ7R infection induces higher levels of IL-1β production with lower pathogenicity in pigs (Figs 5, 9 and 10). Therefore, our systematic study uncovers the mechanisms by which ASFV evades host antiviral inflammatory responses.

IFN is a key player in the host to fight against virus infection, which leads to the establishment of an anti-viral state in nearby cells. The cGAS-STING pathway is critical to mediate the host innate immune responses against invading DNA viruses and is therefore targeted by several DNA viruses-encoded proteins [41]. A precious study showed that ASFV induces type I

IFNs production through the cGAS-STING pathway [18]. To escape the host innate immune responses, ASFV encodes a number of proteins to inhibit type I IFNs production by antagonizing the cGAS-STING signaling. Among them, several members of pMGF360 and pMGF505 have been proved to play a crucial role in determining macrophage host range and interferon response [25,42]. The highly virulent strain Pr4 with MGF360 and MGF505 deletion (Pr4Δ35) induces higher type I IFN production in macrophages compared with its parental virus [25]. For examples, pMGF360-15R/pA276R, a member of MGF360, inhibits the production of IFNs in a NF-κB-independent manner and is restricted to the activation of IRF3 [43]. The pMGF360-15R/pA276R also inhibits type I IFN signaling, and it functions at the level of IRF3 [43]. The pMGF505-7R/pA528R from MGF505 inhibits both IRF3 and ISRE promoter activation [43]. During the manuscript was being reviewed, ASFV CN/GS/2018 strain pMGF505-7R was found to promote ULK1-mediated STING degradation to negatively regulate IFN production [22]. In this study, we also found that nearly all members of pMGFs can inhibit IFN-β production, and pMGF505-7R shows the strongest inhibitory effect (Fig 3B). In addition, we found that pMGF505-7R is able to suppress the promoter activities of IFN-β, IFN-α, ISRE and ISG56, showing a strong inhibition of type I IFN signaling (Fig 4C–4F). Mechanistically, we demonstrated that pMGF505-7R inhibits type I IFN production by binding to IRF3 and blocking its nuclear translocation (Fig 8). Therefore, our results reveal another mechanism by which ASFV escapes from host type I IFN signaling by directly targeting IRF3.

Lack of efficient vaccines against ASF makes it extremely difficult to control and eradicate the disease. Deletion of specific genes using recombination technologies [44] or the CRISPR-Cas9 system [45] opens the possibility of obtaining safer recombinant live attenuated viruses (LAVs). Up to now, a few genes, including DP71L (NL), B119L (9GL), DP96R (UK), DP148R, MGF360 and MGF505 have been identified as critical determinants of the virulence in different ASFV strains [46]. Interestingly, a recent study demonstrated the safety and efficacy of a new recombinant LAV obtained by deleting the MGF360/530 and CD2v virulence factors [47]. In addition, the deletion of the previously uncharacterized I177L gene also provokes the dramatic attenuation of the ASFV and confers sterilizing protection against the virus currently threatening our pig industry [48]. These results indicate that deletion of key virulence gene proves to be a potential strategy for developing novel LAV candidate vaccine. In the study, a recombinant ASFV-Δ7R was generated, and pigs were challenged intramuscularly (i.m.) with ASFV-Δ7R or ASFV-WT at a dose of $10^3$ or $10^5$ HAD$_{50}$ (Fig 9). The body temperatures of all animals increased gradually, peaked at 41.7˚C by day 9 (Fig 9A). Interestingly, pigs challenged with ASFV-WT all died within 10 days, while over 60% of pigs challenged with ASFV-Δ7R survived (Fig 9B), suggesting that ASFV-Δ7R was dramatically attenuated. A recent study also shows that ASFV CN/GS/2018 strain deficient of the MGF505-7R gene was dramatically attenuated and is not lethal to pigs when challenged at a dose of 10 HAD$_{50}$ [22]. While our results suggest that infection of ASFV-Δ7R with high doses are still lethal to piglets and can lead to about 40% of lethality. It is noteworthy that the viral DNA copies in the ASFV-Δ7R-inoculated pigs decreased 2 to 3 orders of magnitude compared with that of the ASFV-WT-inoculated pigs (Fig 9C), although no difference in genome copies between ASFV-WT and ASFV-Δ7R had been found in PAMs *in vitro*. This inconsistence is likely due to involvement of other immune cells in addition to PAMs, such as T cells and B cells, which function cooperatively in the swine body to suppress attenuated ASFV-Δ7R replication while PAMs alone are not sufficient to support the suppression.

In summary, in this study we found that ASFV infection induces low levels of IL-1β and type I IFNs, and the viral pMGF505-7R plays critical role in suppressing the production of IL-1β and type I IFNs. Mechanistically, pMGF505-7R binds to IKK complex to inhibit TLRs/MyD88-dependent pro-IL-1β transcription and interacts with NLRP3 to inhibit NLRP3

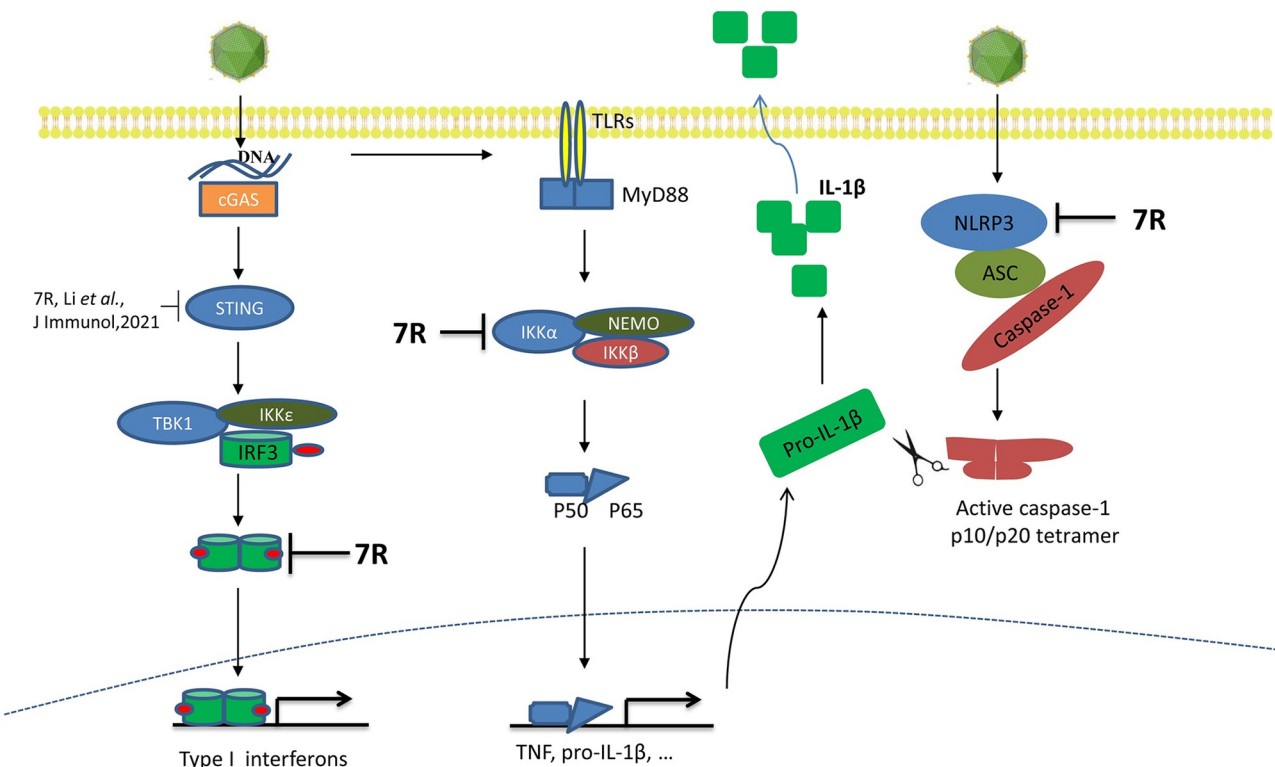

**Fig 11. Schematic model of pMGF505-7R inhibits IL-1β and type I IFN production.** After ASFV infection, pMGF505-7R inhibits IL-1β and IFN-β production. Mechanistically, pMGF505-7R interacts with IKK complex to inhibit NF-κB activation and binds to NLRP3 to inhibit inflammasome formation, leading to decrease IL-1β production. Moreover, pMGF505-7R inhibits the nuclear translocation of IRF3 to block type I IFN production.

inflammasome assembly (Fig 11). Moreover, we demonstrate that pMGF505-7R binds to IRF3 to block its nuclear translocation and thus suppresses type I IFN signaling. Our findings help understand the functions of ASFV-encoded pMGF505-7R and its role in viral infection, which might help design antiviral agents or live attenuated vaccines to control ASF.

## Materials and methods

### Ethics statements

All experiments with ASFV HLJ/18, ASFV-EGFP-7R or ASFV-Δ7R were conducted within the enhanced biosafety level 3 (P3+) and level 4 (P4) facilities in the Harbin Veterinary Research Institute (HVRI) of the Chinese Academy of Agricultural Sciences (CAAS) approved by the Ministry of Agriculture and Rural Affairs. This study was carried out in strict accordance with the recommendations in the Guide for the Care and Use of Laboratory Animals of the Ministry of Science and Technology of the People's Republic of China.

### Cells and viruses

PAMs were isolated from the lung lavage fluid of 4-week-old healthy specific pathogen free (SPF) piglets as previously described [49] and maintained in RPMI-1640 medium containing 10% fetal bovine serum (FBS, Hyclone), 100 U/ml penicillin, 50 mg/ml streptomycine and non-essential amino acid (NEAA, Gibco). The PAM cell line 3D4/21 (CRL-2843) established by transformation of PAMs with SV40 large T antigen [50] was purchased from the American

Type Culture Collection (ATCC) and maintained in RPMI-1640 medium supplemented with 10% FBS. HEK293T cells obtained from ATCC were cultured in Dulbecco's Modified Eagle's medium (DMEM) supplemented with 10% FBS. All the cells were maintained at 37˚C with 5% CO$_2$. ASFV (HLJ/18 strain, GenBank accession number: MK333180.1) was isolated from pig as previously described [1]. HSV-1 and Sendai virus (SeV) were kindly provided by Prof. Hongbin Shu (Wuhan University, China).

## Antibodies and reagents

LPS, Nigericin, mouse anti-HA monoclonal antibody (mAb), rabbit anti-HA polyclonal antibody (pAb), mouse anti-Flag mAb and rabbit anti-Flag pAb were purchased from Sigma-Aldrich. Poly(dA:dT), TNF-α, mouse anti-IL-1β mAb and mouse anti-glyceraldehyde-3-phosphate dehydrogenase (GAPDH) pAb were purchased from Thermo. Rabbit anti-NLRP3 pAb, anti-caspase-1 p20 pAb and anti-swine ASC mAb were prepared by immunizing the rabbits with recombinant proteins [49]. Goat anti-rabbit and anti-mouse secondary antibodies conjugated to IRDye 800CW and IRDye 680LT used for immunoblotting were purchased from LI-COR.

## Stimulation of PAMs and detection of inflammatory cytokines

For ASFV infection, 1×10$^6$ PAMs were infected with at an MOI of 0.01, 0.1, and 1.0 ASFV for 2 h at 37˚C and then washed twice with PBS and the cells were cultured in indicated medium. The cells were treated with LPS (100 ng/ml) for 8 h and then stimulated with Nigericin (5 μM) for another 4 h, or with LPS (100 ng/ml) and poly(dA:dT) (5 μg/ml) for 12 h, or with SeV (1 MOI) for 12 h, or poly(dA:dT) (5 μg/ml) for 12 h. The cell supernatants were collected to measure the concentrations of several cytokines such as IL-1β, IFN-α and IFN-β by Enzyme-Linked Immunosorbent Assay (ELISA) kit (Ray Biotech, Norcross, GA). The cells were harvested for qPCR analysis. PAMs were infected with HSV-1 as positive control, while the uninfected PAMs were used as negative control.

## iGLuc-based inflammasome reconstruction system

iGLuc-based inflammasome reconstruction system were constructed based on *Gaussia* luciferase with high sensitivity and specificity as previously described [51]. Briefly, HEK293T cells were plated in 48-well plates at 5×10$^4$ per well and then transiently transfected with a total amount of 500 ng of DNA per well using 1 μl of X-tremeGENE HP DNA Transfection Reagent (Roche). The 500 ng of DNA contains iGLuc-based NLRP3 inflammasome system (100 ng iGLuc, 10 ng pCAGGS-caspase-1,10 ng pCAGGS-ASC, and 12.5 ng pCAGGS-NLRP3), the a plasmid containing a tested gene and empty vector. After 24 h, the supernatants were collected for GLuc readout.

## Transfection and luciferase reporter assays

Reporter assay was performed as described previously [52]. Briefly, HEK293T cells were transfected with plasmids as indicated, in combination with 100 ng of a reporter plasmid and 5 ng of the pRL-TK plasmid. In each experiment, the total amount of DNA (500 ng) in each sample was kept constant by supplementation with the empty vector. At 24 h post transfection (hpt), the activities of Firefly and Renilla luciferase were determined using the Dual-Luciferase Reporter Assay System (Promega Corporation) according to the manufacturer's instructions.

## siRNA knockdown

The targeting sequences of siRNAs are listed in S1 Table. The transfection of siRNAs was performed with HiPerFect Transfection Reagent (QIAGEN) by following the manufacturer's instructions. 48 h after siRNA transfection, the cells were infected with ASFV at an MOI of 1.0 for 24 h. siRNA knockdown efficiency of the target protein was assessed by qPCR or Western blot. The concentrations of IL-1β were measured by qPCR or ELISA.

## Quantitative PCR (qPCR)

Gene expression levels of IL-1β and other indicated genes were tested by real-time RT-PCR using SYBR Green detection system. The primers are listed in S1 Table. The expression levels of these genes were normalized to GAPDH or HPRT gene. The final mRNA levels of these genes in this study were normalized using the comparative cycle threshold method.

For ASFV genome DNA copy detection, ASFV genomic DNA was extracted using DNA Mini Kit (Qiagen, Germany) from cells, tissue homogenates or EDTA-treated whole peripheral blood. qPCR was carried out on a QuantStudio5 system (Applied Biosystems, USA) according to the OIE-recommended procedure [53].

## Confocal microscopy

Cells were fixed with 4% paraformaldehyde and permeabilized with 0.1% Triton-X 100. After blocking with 10% FBS, cells were incubated with anti-Flag and anti-HA antibodies for 1 h. After washing with 1× cold PBS (pH7.4) for three times, the cells were incubated with indicated secondary antibodies. Samples were visualized with a Leica SP2 confocal system (Leica Microsystems, Germany).

## Nuclear and Cytoplasmic Extraction

For ASFV infection, $1 \times 10^6$ PAMs were infected with ASFV for 2 h at 37˚C and then washed twice with PBS and the cells were cultured in indicated medium. The cells were treated with TNF-α (15 ng/mL) or SeV (MOI = 1) stimulation for 24 h. The cells were collected and treated with NE-PER Nuclear and Cytoplasmic Extraction Reagents (Thermo Scientific). The nuclear and cytoplasmic compartments were detected by Western blotting.

## Co-immunoprecipitation (Co-IP) and Western blotting

HEK293T cells were transfected with plasmids expressing indicated proteins and Flag-MGF505-7R using Lipofectamine 2000. The total amount of DNA in each sample was kept constant by supplementation with the empty vector with the tag as a control. The cells were collected at 36–48 hpt and lysed in cell lysis buffer (50 mM Tris, pH 7.4, 150 mM NaCl, 1% Triton-X 100, 5 mM EDTA, and 10% glycerol). The lysates were immunoprecipitated by anti-Flag (M2) agarose beads (Sigma). For endogenous protein interaction, PAMs mock-infected or infected with ASFV-EGFP-7R (MOI = 1) for different time points were washed, lysed in lysis buffer, clarified by centrifugation for 30 min at 4˚C and normalized to equal amounts of total protein. The lysates were incubated with indicated antibodies or IgG for 8 h at 4˚C, and the proteins were captured using 10 μl of protein A+G-Sepharose. The pellets were resolved by 10–12% SDS-PAGE gel and subjected to Western blotting. Equal amount of proteins from whole cells lysates were subjected to Western blotting and used as the loading control.

## ASC speck and oligomerization

CRL-2843 cells were transfected with plasmids encoding GFP-ASC and pMGF-505-7R. At 24 hpt, the cells were stimulated with Nigericin (5 μM) for 1 h, and then observed for the ASC specks with a Leica SP2 confocal system. HEK293T cells were transfected with different combinations of plasmids encoding GFP-ASC, NLRP3 and pMGF-505-7R. At 24 hpt, the cells were harvested, washed in PBS with 1% FCS and fixed in 2% PFA for 30 min at 4˚C. The percentage of EGFP-high (EGFP-H) population in the mean fluorescence intensities of EGFP (EGFP+) population was determined by flow cytometry. PAMs were infected with ASFV-Δ7R or ASFV-WT for 24 h and then lysed. The pellets were washed with PBS for three times and cross-linked using fresh DSS (2 mM, Sigma) at 37˚C for 30 min. Then the cross-linked pellets were centrifuged and mixed with SDS-loading buffer for western blot analysis.

## Generation of MGF505-7R gene-deleted ASFV (ASFV-Δ7R)

A recombinant ASFV was generated by CRISPR-Cas9 and homologous recombination in PAMs. Plasmid pBluscript II SK (-) was used as a backbone, a reporter gene cassette with the EGFP gene with the ASFV p72 late gene promoter was inserted at the E*co*RV restriction site. Recombinant transfer vector (p72-EGFPΔMGF505-7R) containing 1000 bp upstream homologous arm of MGF505-7R, a reporter gene cassette, followed by 1000 bp downstream homologous arm of MGF505-7R. The primers were used in S1 Table. The two NLS within the Cas9 ORF of vector pX330 were removed and the overlapping complementary oligonucleotide primers sg7R-F and sg7R-F (S1 Table) were hybridized and inserted into the B*bs*I-digested vector, resulting in pX330-sg7R. PAMs were transfected with linearized transfer vector and pX330-sg7R using Fugene HD following the manufactures protocol. At 24 hpt, the cells were infected with ASFV (MOI = 1) for another 24 h. Cells with EGFP expression were screened out and the recombinant virus was selected after 10 rounds of plaque purification in PAMs based on GFP expression. The DNA sequence covering the modified region was amplified and sequenced.

## Generation of a recombinant ASFV with EGFP-tagged pMGF505-7R (ASFV-EGFP-7R)

The homologous arms and sgRNA are the same as generation of ASFV-Δ7R. First, EGFP and MGF505-7R fusion gene (EGFP-7R) were constructed. And then, a new recombinant transfer vector was constructed to replace reporter gene cassette (p72-EGFP) in p72-EGFPΔMGF505-7R vector with EGFP-7R. PAMs were transfected with linearized transfer vector and pX330-sg7R using Fugene HD following the manufactures protocol. 24 h after transfection, cells were infected with ASFV (MOI = 1) for another 24 h. Cells with EGFP expression were screened out and the recombinant virus was selected after 10 rounds of plaque purification in PAMs based on GFP expression. The DNA sequence covering the modified region was amplified and sequenced.

## Detection of the transcriptome of PAMs infected with ASFV-WT or ASFV-Δ7R

PAMs were mock infected, infected with ASFV-WT or infected with ASFV-Δ7R at an MOI of 1. At 24 hpi, the cells were washed once with PBS before adding TRIzol (Invitrogen). The samples were sent to the company for sequencing (ANOROAD, Beijing, China). Briefly, purified total RNAs from ASFV-WT and ASFV-Δ7R-infected cells were treated with DNase I (Takara) followed by column purification (RNeasy MinElute Cleanup Kit [QIAGEN]) and used for the

experiments. The fragment buffer was added to the mRNA to make the fragment into a short fragment. Reverse transcription was done with random primers, and the library was constructed. The qualified library was sequenced by Illumina platform. Clean reads were acquired from raw data by the FASTQ_Quality_Filter tool of the FASTX-tool kit, and then mapped to the African swine fever virus reference genome (MK333180.1). The unique mapped reads were used for the further analysis. Gene expression levels were normalized using the FPKM method [54]. Differentially expressed genes (DEGs) were analyzed by the edgR using fold change$\geq$2 and FDR<0.05 thresholds [55].

## The virulence of ASFV-Δ7R in domestic pigs

Animal experiments were performed within the animal biosafety level 4 facilities at HVRI following a protocol approved by the Animal Ethics Committee of HVRI of CAAS and the Animal Ethics Committee of Heilongjiang Province, China. Eighteen 4-week-old healthy specific pathogen-free piglets were randomly assigned into four groups (7 piglets inoculated with ASFV-Δ7R, $10^3$ HAD$_{50}$/piglet (1 mL); 7 piglets inoculated with ASFV-Δ7R, $10^5$ HAD$_{50}$/piglet (1 mL); 4 piglets inoculated with ASFV-WT, $10^3$ HAD$_{50}$/piglet (1 mL); 4 piglets inoculated with PBS). The piglets were monitored daily for clinical signs prior to feeding, including anorexia, lethargy, fever, and emaciation. The serum samples were collected at days 0, 1, 5, 8 and 10 post infection (dpi) for inflammatory cytokines and virus load detection. ASFV-infected piglets were euthanized in the moribund stage. At 19 dpi, all surviving piglets were euthanized. The tissue samples from heart, lung, spleen, tonsil, thymus and five lymph nodes (intestinal lymph node, inguinal lymph node, submaxillary lymph node, bronchial lymph node, and gastrohepatic lymph node) were collected for ASFV detection. The amounts of IL-1β, IFN-β, IFN-α in the serum isolated from piglets were determined by the enzyme-linked immunosorbent assay (ELISA) kits purchased from R&D Systems.

## Statistical analysis

Data were analyzed for statistical significance by two-tailed student's *t-test*.

## Supporting information

**S1 Fig. Detection of ASFV pMGFs expressions by IFA and WB using anti-Flag antibody. (A-B)** HEK293T cells were transfected with plasmids expressing ASFV-encoded pMGFs in the presence of the iGLuc-based NLRP3 inflammasome system or IFN-β promoter reporter. The cells were probed with mouse anti-Flag mAb, and then observed by microscopy **(A)**. The cells were detected by Western blot using anti-Flag mAb **(B)**. * means a nonspecific band.
(TIF)

**S2 Fig. Generation of ASFV with deletion of MGF505-7R gene (ASFV-7ΔR). (A)** Schematic representation of generation of ASFV-Δ7R. The MGF505-7R gene segment was replaced with the p72-EGFP reporter gene cassette. **(B)** PAMs were infected with ASFV (ASFV-WT) or ASFV-Δ7R. At 24 hpi, the cells were observed by microscope. **(C)** Agarose gel (1%) showing the result of the conventional PCR to amplify of the genomic segment containing the targeted gene. **(D)** Growth kinetics in PAMs for ASFV-Δ7R and ASFV-WT. PAMs were infected with ASFV-Δ7R or ASFV-WT (MOI = 0.01), and samples were taken from three independent experiments at the indicated time points and titrated.
(TIF)

**S3 Fig. Viral encoded-genes transcription in ASFV-Δ7R-infected PAMs compared with that of ASFV-WT-infected PAMs. (A)** Detection of RNA SEQ correlation among samples.

**(B)** The heatmap of different categories in diverse pathological classes based on TCM Diagnostic System. Each row refers to a sample. The color variety showed the frequency of the targets in each category. **(C)** Gene expression levels were normalized using the FPKM method. Differentially expressed genes (DEGs) were analyzed by the edgR using fold change ≥ 2 and FDR < 0.05 thresholds.
(TIF)

**S4 Fig. ASFV-Δ7R infection induces IL-1β transcription through the TLRs/MyD88 pathway in PAMs. (A-B)** PAMs were transfected with control siRNA (siNC) or siRNAs targeting each of the TLRs or MyD88. At 24 hpt, the cells were infected with ASFV-Δ7R at an MOI of 1 for another 24 h, then the mRNA levels of IL-1β were detected by qPCR **(A)**. The mRNA levels of TLRs and MyD88 were detected to confirm the knockdown efficiencies mediated by the siRNAs **(B)**. A $p$ value of less than 0.05 was considered statistically significant. $^*p < 0.05$, $^{**}p < 0.01$, $^{***}p < 0.001$.
(TIF)

**S5 Fig. Generation of recombinant ASFV with EGFP-tagged pMGF505-7R expression (ASFV-EGFP-7R). (A)** Schematic representation of generation of ASFV with EGFP-tagged pMGF505-7R. The EGFP was inserted N-terminal of MGF505-7R. **(B)** PAMs were infected with ASFV (ASFV-WT) or ASFV-EGFP-7R. At 24 hpi, the cells were observed by microscope. **(C)** Agarose gel (1%) showing the result of the conventional PCR to amplify of the genomic segment containing the targeted gene. **(D)** PAMs were mock-infected or infected with ASFV-EGFP-7R (MOI = 1) for 36 h, and the expressing levels of EGFP-7R were detected by Western blotting. **(E)** Growth kinetics in PAMs for ASFV-EGFP-7R and ASFV-WT. PAMs were infected with ASFV-Δ7R or ASFV-WT (MOI = 0.01), and samples were taken from three independent experiments at the indicated time points and titrated. **(F)** HEK293T cells were transfected with increasing doses of a plasmid expressing pEGFP-MGF505-7R in the presence of the iGLuc-based NLRP3 inflammasome system, and the supernatants were assessed 24 hpt for luciferase activity. **(G)** HEK293T cells were co-transfected with increasing doses of a plasmid expressing EGFP-MGF505-7R and IFN-β promoter reporter for 24 h and then the cells were stimulated with poly(I:C) (1 μg) for another 12 h, the cells were then collected, and the luciferase activities were measured. A p value of less than 0.05 was considered statistically significant. $^*p < 0.05$, $^{**}p < 0.01$, $^{***}p < 0.001$.
(TIF)

**S6 Fig. ASFV-Δ7R infection induces IL-1β dependent on NLRP3 in PAMs. (A-B)** PAMs were transfected with siRNAs targeting NLRP3. At 24 hpt, the cells were infected with ASFV-Δ7R at an MOI of 1 for another 24 h, then the mRNA levels of NLRP3 were detected by qPCR **(A)**, the protein levels of NLRP3 were detected by Western blot and the secretion of IL-1β were detected by ELISA **(B)**. A $p$ value of less than 0.05 was considered statistically significant. $^*p < 0.05$, $^{**}p < 0.01$, $^{***}p < 0.001$.
(TIF)

**S7 Fig. ASFV-Δ7R infection induces NLRP3 inflammasome activation in PAMs. (A)** Flag-pMGF505-7R and HA-NLRP3 were expressed individually or together in HEK293T cells. The cells were probed with rabbit anti-HA mAb and mouse anti-Flag mAb, stained with nucleus marker DAPI and then observed by confocal microscopy. **(B)** HEK293T cells were transfected with a plasmid encoding Flag-MGF505-7R alone or together with a plasmid encoding HA-NLRP3 or its truncated mutants. 48 hpt, cell lysates were collected and immunoprecipitated with anti-Flag antibody, followed by immunoblotting for HA-tagged NLRP3 and its truncated mutants. **(C)** HEK293T cells were transfected with plasmids expressing GFP-ASC

alone, or GFP-ASC and HA-NLRP3, or GFP-ASC and HA-NLRP3 together with increasing amounts of a plasmid expressing Flag-pMGF505-7R. 24 hours post transfection, cells were harvested, fixed, and the fraction of cells containing ASC specks was quantified by flow cytometry.
(TIF)

**S1 Table. Primers, siRNA and sgRNA used in this study.**
(DOCX)

**S2 Table. The DNA sequence covering the modified region of ASFV.**
(DOCX)

## Author Contributions

**Conceptualization:** Jiangnan Li, Zhigao Bu, Changjiang Weng.

**Data curation:** Jiangnan Li, Jie Song.

**Formal analysis:** Jiangnan Li, Jie Song, Zhigao Bu, Changjiang Weng.

**Funding acquisition:** Jiangnan Li, Changjiang Weng.

**Investigation:** Jiangnan Li, Jie Song, Li Kang, Li Huang, Shijun Zhou.

**Methodology:** Jiangnan Li, Jie Song, Li Kang, Li Huang, Shijun Zhou, Xianfeng Zhang, Xijun He.

**Project administration:** Jiangnan Li, Changjiang Weng.

**Resources:** Jiangnan Li, Jie Song, Li Kang, Li Huang, Shijun Zhou, Dongming Zhao, Zhigao Bu.

**Supervision:** Zhigao Bu, Changjiang Weng.

**Validation:** Li Huang, Jun Zheng, Changyao Li.

**Writing – original draft:** Jiangnan Li.

**Writing – review & editing:** Jiangnan Li, Jie Song, Liang Hu, Zhigao Bu, Changjiang Weng.

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
