## [Decision Letter · Decision Letter 0]

1 Apr 2021

Dear Mr. Weng,

Thank you very much for submitting your manuscript "pMGF505-7R determines pathogencity of African swine fever virus infection by inhibiting IL-1β and type I IFN production" for consideration at PLOS Pathogens. As with all papers reviewed by the journal, your manuscript was reviewed by members of the editorial board and by several independent reviewers. In light of the reviews (below this email), we would like to invite the resubmission of a significantly-revised version that takes into account the reviewers' comments.

The reviewers appreciated the attention to an important problem but raised some substantial concerns about the manuscript as it currently stands. These issues must be addressed before we would be willing to consider a revised version of your study. We cannot of course promise publication at that time. 

Your revision should address the specific points made by each reviewer.

Two of the reviewers point out the similarity of your results to a recently published paper Li et al. (doi/10.4049/jimmunol.2001110). In any revision you should make clear novelty of your data beyond those previously published.

We cannot make any decision about publication until we have seen the revised manuscript and your response to the reviewers' comments. Your revised manuscript is also likely to be sent to reviewers for further evaluation.

Sincerely,

Linda Kathleen Dixon

Guest Editor

PLOS Pathogens

Adolfo García-Sastre

Section Editor

PLOS Pathogens

Kasturi Haldar

Editor-in-Chief

PLOS Pathogens

orcid.org/0000-0001-5065-158X

Michael Malim

Editor-in-Chief

PLOS Pathogens

orcid.org/0000-0002-7699-2064

Reviewer's Responses to Questions

**Part I - Summary**

Reviewer #1: The manuscript by Li et al., contributes interesting data about the determination on how ASFV infection impaired the immune response by inducing low levels of IL-1b and type I IFNs. It also identifies some of the ASFV genes that are responsible for the modulation of the immune response during the ASFV infection. Authors demonstrate this modulation by doing different experiments in vitro, by using PAMs, and in vivo on pigs. This contribution on the field of ASFV research is important not only to better understand the pathogenesis of ASFV but also in the study of potential vaccine candidates.

Overall, the paper has good quality for publication.

The hypothesis in the paper have been mostly answered by experimental procedures which confirm the hypothesis. The methodology and results are well described. The results are clear and answered the topic studied. The paper is well written and the bibliography is up to date.

As a suggestion, it could have been interesting to study the consequences of the IKK alpha interaction downstream of the interaction. It could be interesting to analyze p65 translocation or the phosphorylation of IKB alpha.

The research topic is well described and the methodology is well planned from the beginning (cell culture) to the end of the study (animal models). Finally, all results are accompanied by statistical data.

However, there are some precisions and additions that should be done:

- Luciferase assays should clarify if authors include empty vector with just the tag (MGF505-7R) as a control. If not, authors should include it necessarily.

- Immunoprecipitation assays should include the empty vector with the tag as a control.

- In the screening to select optimal MGF some controls are missing. This reviewer find that the Supplementary IFI are not the optimal approach to test if expression was similar with all the constructions.

It is necessary to include a robust analysis of the expression levels between plasmids given the very close levels of luminescence inhibition found.

- More level of detail would be desirable for Materials and Methods section. Concentrations of plasmids or stimulators are not easy to find. We suggest to include a Table in M&M with all plasmids and concentrations used.

Reviewer #2: The manuscript describes some new characteristics of ASFV MGF505 members in modulating IL-1β and type I IFN production in infected cells and pigs. Among the ASFV MGF505 members, the authors have found the strongest inhibitory effect of MGF505-7R gene indicating its potential in developing a new attenuated vaccine against ASF. Subsequently, the authors have studied signaling pathways and localization of the ASFV MGF505-7R. They have shown that MGF505-7R mode of action is primarily via by the interaction with IKKα complex which leads to NF-κB inhibition and bounding to NLRP3, leading to decreased IL-1β production.

It is worth noting that several studies have already described similar data regarding the role of ASFV MGF505 in IFN inhibition. In particular, Dan Li et al (2021) have presented identical results using recombinant ASFV-Δ7R (African Swine Fever Virus MGF-505-7R Negatively Regulates cGAS–STING-Mediated Signaling Pathway).

Remarkable results have been generated using recombinant ASFV lacking the MGF505-7R gene (ASFV-Δ7R) and protective potential of ASFV-Δ7R has been discovered (60% of animals survived the challenge).

Reviewer #3: The authors study the role of MGF505-7R in modulating IL-1� and type I IFN response in porcine alveolar macrophages (PAMs) and during infection in swine. Authors showed the pattern of cytokines expression (by ELISA and qRT-PCR) in PAMs infected with ASFV. They then demonstrated the inhibitory effect of ASFV infection in the IL-1� and type I IFN response in PAMs treated with different inducers (as LPS and polydA:dT). Following this, they showed the inhibitory effect of MGF505-7R in IL-1� and type I IFN production in HEK293T cells transfected with 15 different ASFV MGF genes and the corresponding reporter systems. MGF505-7R was then shown to inhibit NF-�B, and IFN�/� activation in a dose fashion in HEK293T cells transfected with the corresponding genes and reporter systems. A virus lacking MGF505-7R gene (ASFV-�7R) induced higher levels of IL-1� and IFN� when infecting PAMs than the wild type (WT) ASFV does. Following this, siRNA specific for TLRs inhibit the increased levels of IL-1� in ASFV-�7R infected PAMs. Also, MGF505-7R over expression inhibits NF-�B activation provoked by LPS, IKK� or IKK�, and IKK�, and MGF505-7R co-precipitate with IKK� when constructs encoding them were co-transfected. Also, PAMs transfected with siRNA specific for NLRP3 gene decreased the IL-1� expression during ASFV-�7R infection and co-precipitation of MGF505-7R and NLPR3 were shown to occurs in HEK293T cells transfected with both genes. In addition, over expression of MGF505-7R inhibit IFN� activation promoted by different inducers (including IRF3), and cells co-transfected with the corresponding genes were used to co-precipitate MGF505-7R and IRF3. In addition, nucleus translocation of IRF3 is halted by expression of MGF505-7R. ASFV-�7R showed an attenuated phenotype compared to the WT virus. ASFV-�7R infected animals presented a delayed presentation of the disease, at 60% survival, and significantly less virus genome than those infected with the WT virus. In addition, ASFV-�7R infected animals presented decreased levels of IL-1� and IFN�.

The manuscript provides evidence of the function of ASFV gene MGF505-7R indicating its role in IL-1� and IFN� down expression, describing mechanisms for the modulation of the expression of both genes and showing the involvement of MGF505-71R gene in virulence during pig infection.

The importance of ASFV as an etiological agent of the enormous pandemic currently affecting pig production in Eurasia make the study of ASFV genes important towards the generation of information that may allow for the development of effective virus countermeasures.

ASFV MGF genes have been already described as clearly responsible for down regulating the innate immune response (particularly type-I IFN) during the infection in pigs (reviewed in ref. 4). Several virus genes belonging to the MGFs have been implicated in the down modulation of the host immune response, particularly type I IFN (refs 18, 19, 21, 24, doi.org/10.3390/v13020255).

Very importantly for this manuscript, Li et al. (doi/10.4049/jimmunol.2001110) recently demonstrated the role of the same gene, MGF505-7R, in down modulating host IFN� response and involved in the process of virus virulence during the infection in pigs. As in the manuscript under review, manuscript Li et al., described a mechanism of how MGF505-7R down regulates IFN� response in vitro cell cultures and how an ASFV lacking MGF505-7R gene showed an attenuated phenotype in pigs inducing higher host IFN� response than the WT virus does. Differences reside in that the current report presents the case that MGF505-7R targets specific adaptor proteins (TLRs, NF-kB and NLRP3) of the innate immunity signaling pathway to inhibit the expression of IL-1� and type I IFN�/� while Li et al., indicated that MGF505-7R targets the cGAS-STING cystosolic DNA sensing pathway which function upstream from NF-kB activation. Both reports make the case that MGF505-7R regulates different sensing and adaptor proteins that ultimately lead to the production of IFN. Also, using different methodologies both reports demonstrated that overexpression of MGF505-7R inhibits the translocation of IRF-3 to the nucleus.

In my opinion, the strong parallelism between results already published by Li et al., and the manuscript under review severely affects the novelty of the results reported in the later.

Some specific points:

1- Lines 136-163: these “introductory” results showing that ASFV infection alter the expression of cytokines in PAMs could be deleted since they are not novel nor have incidence in the results presented later.

2- Lines 199-208: ASFV-�7R is an essential tool in this manuscript therefore the development of recombinant ASFV-�7R should be better described. Was the MGF505-7R gene partially/completely deleted? May the deletion compromise adjacent genes? Authors should report the full length sequence of virus genome to ensure the absence of unwanted mutations elsewhere in the genome.

3- Lines 241-242: authors indicate that MGF505-7R expression inhibits NF-�B activation provoked by IKK� in a dose dependent manner. Figure 6B does not show a dose dependent response. The same is true regarding the inhibitory effect of MGF505-7R expression on the activation of IFN� promoter by IRF3-5D (lines 290-298 and Figure 8C).

4- All immune co-precipitations presented in the manuscript are made in target cells transfected with plasmids encoding the proteins under study. Usually, levels of protein expression achieved by this procedure are not physiological; therefore, it may not represent what actually happens in the infected cells. Authors should mention this issue and, also, try to confirm that the reported interaction between proteins indeed occurs during the ASFV infection.

5- Lines 301-306: confocal studies showing the effect of MGF505-7R on IRF3 translocation in PAMs must be confirmed by comparing the differential presence of IRF3 translocation in cells infected with ASFV WT or ASFV-�7R.

6- Lines 325-342: determinations described here probably were performed in the same group of animals already described in lines 307-324. If they are the same animals, are panel C in Figure 9 and panel D in Figure 10 showing the same results?

**Part II – Major Issues: Key Experiments Required for Acceptance**

Reviewer #1: There are some precisions and additions that should be done:

- Luciferase assays should clarify if authors include empty vector with just the tag (MGF505-7R) as a control. If not, authors should include it necessarily.

- Immunoprecipitation assays should include the empty vector with the tag as a control.

- In the screening to select optimal MGF some controls are missing. This reviewer find that the Supplementary IFI are not the optimal approach to test if expression was similar with all the constructions.

It is necessary to include a robust analysis of the expression levels between plasmids given the very close levels of luminescence inhibition found.

- More level of detail would be desirable for Materials and Methods section. Concentrations of plasmids or stimulators are not easy to find. We suggest to include a Table in M&M with all plasmids and concentrations used.

Reviewer #2: Several studies have shown that the ASFV has an arsenal of immunomodulatory proteins and one of the most sophisticated system of immune evasion among viruses. In order to decipher such complex interactions between cell and virus proteins, single virus protein analysis unable to reveal the whole specter of multimodal actions that likely to happen in infected cells or organism.

1. All MGF505 members have shown an inhibitory effect on IL-1β production. However, the authors have chosen just one protein (MGF505-7R) with the strongest potential. The would be interesting to see whether the MGF505 proteins have a complementary effect on the IL-1β and type I IFN production.

2. The authors claim that 60% of pigs infected with the ASFV-Δ7R survived the challenge. I would recommend to add the comparative data between survived animals and those who succumbed the disease (40%) in the context of IL-1β and type I IFN level.

Reviewer #3: 4- All immune co-precipitations presented in the manuscript are made in target cells transfected with plasmids encoding the proteins under study. Usually, levels of protein expression achieved by this procedure are not physiological; therefore, it may not represent what actually happens in the infected cells. Authors should mention this issue and, also, try to confirm that the reported interaction between proteins indeed occurs during the ASFV infection.

5- Lines 301-306: confocal studies showing the effect of MGF505-7R on IRF3 translocation in PAMs must be confirmed by comparing the differential presence of IRF3 translocation in cells infected with ASFV WT or ASFV-�7R.

**Part III – Minor Issues: Editorial and Data Presentation Modifications**

Reviewer #1: As minor revisions, please check figure 5 A and B where I suggest include the name of each condition in the experiment under each column (similar to Figure 5 C and D for clarification).

Reviewer #2: -Line 76 - ASF is arguably the most dangerous swine disease, but not the most serious viral diseases as stated in the manuscript. Please attenuate the statement.

-Line 86 - Please edit lower case letters "innate"

-Line 112- Please edit a capitalized word "Pathway"

-Line 313 - please rephrase the sentence.

-Fig.1 - I assume that ASFV-WT has been used here. Please use the same virus title along the manuscript.

- MOI 0.5 of ASFV has been used for animal trial, however MOI 0.01, 0.1 and 1 have been tested in vitro. Please provide some rationale for this.

- Figure. 5C - No difference in genome copies between ASFV-WT and ASFV -d7R ha seen found in infected cells. However, decreased amount of ASFV0d7R genomes has been detected in infected animals. Please elaborate this founding in the discussion section.

- Fig.7. C - is hard to interpret and need more clarification.

- Fig.11 describes the signaling pathway and a role of MGF-7R in it. The figure will benefit if other viral proteins involved in the same pathway will be noted.

Reviewer #3: None

PLOS authors have the option to publish the peer review history of their article (what does this mean?). If published, this will include your full peer review and any attached files.

Reviewer #1: No

Reviewer #2: No

Reviewer #3: No
---

## [Editor Report · Decision Letter 1]

21 Jun 2021

Dear Mr. Weng,

We are pleased to inform you that your manuscript 'pMGF505-7R determines pathogenicity of African swine fever virus infection by inhibiting IL-1β and type I IFN production' has been provisionally accepted for publication in PLOS Pathogens.

Best regards,

Linda Kathleen Dixon

Guest Editor

PLOS Pathogens

Adolfo García-Sastre

Section Editor

PLOS Pathogens

Kasturi Haldar

Editor-in-Chief

PLOS Pathogens

orcid.org/0000-0001-5065-158X

Michael Malim

Editor-in-Chief

PLOS Pathogens

orcid.org/0000-0002-7699-2064
---

## [Editor Report · Acceptance letter]

22 Jul 2021

Dear Mr. Weng,

We are delighted to inform you that your manuscript, "pMGF505-7R determines pathogenicity of African swine fever virus infection by inhibiting IL-1β and type I IFN production," has been formally accepted for publication in PLOS Pathogens.

Best regards,

Kasturi Haldar

Editor-in-Chief

PLOS Pathogens

orcid.org/0000-0001-5065-158X

Michael Malim

Editor-in-Chief

PLOS Pathogens

orcid.org/0000-0002-7699-2064